# Gene expression variation across genetically identical individuals predicts reproductive traits

**Amy K Webster**[1,2], **John H Willis**[1], **Erik Johnson**[1], **Peter Sarkies**[3], **Patrick C Phillips**[1]*

[1]Institute of Ecology and Evolution, University of Oregon, Eugene, United States; [2]Department of Biological Science, Florida State University, Tallahassee, United States; [3]Department of Biochemistry, University of Oxford, Oxford, United Kingdom

## eLife Assessment

This **important** study addresses the role of non-genetic factors in individual differences in phenotype. Using *C. elegans*, the study finds that non-genetic differences in gene expression, partly influenced by the environment, correlate with individual differences in two reproductive traits. This supports the use of gene expression data as a key intermediate for understanding complex traits. The clever study design makes for **compelling** evidence.

*For correspondence:
pphil@uoregon.edu

**Competing interest:** The authors declare that no competing interests exist.

**Abstract** In recent decades, genome-wide association studies (GWAS) have been the major approach to understand the biological basis of individual differences in traits and diseases. However, GWAS approaches have limited predictive power to explain individual differences, particularly for complex traits and diseases in which environmental factors play a substantial role in their etiology. Indeed, individual differences persist even in genetically identical individuals, although fully separating genetic and environmental causation is difficult in most organisms. To understand the basis of individual differences in the absence of genetic differences, we measured two quantitative reproductive traits in 180 genetically identical young adult *Caenorhabditis elegans* roundworms in a shared environment and performed single-individual transcriptomics on each worm. We identified hundreds of genes for which expression variation was strongly associated with reproductive traits, some of which depended on individuals' historical environments and some of which was random. Multiple small sets of genes together were highly predictive of reproductive traits, explaining on average over half and over a quarter of variation in the two traits. We manipulated mRNA levels of predictive genes to identify a set of causal genes, demonstrating the utility of this approach for both prediction and understanding underlying biology. Finally, we found that the chromatin environment of predictive genes was enriched for H3K27 trimethylation, suggesting that gene expression variation may be driven in part by chromatin structure. Together, this work shows that individual, non-genetic differences in gene expression are both highly predictive and causal in shaping reproductive traits.

## Introduction

Over the past two decades, genome-wide association studies (GWAS) have sought to identify the genetic basis of individual differences driving traits and diseases. While there have been some notable successes (*Visscher et al., 2017*; *Edwards et al., 2005*), GWAS analyses often require hundreds of thousands of individuals to identify genetic loci that typically explain a small fraction of variation in traits and diseases (*Nolte et al., 2017*), and the clinical utility of such loci is often unclear (*Alsheikh*

*et al., 2022*; *Gallagher and Chen-Plotkin, 2018*). Complex traits and diseases that are driven by a strong environmental component are particularly problematic for GWAS, as these environmental inputs cannot be experimentally controlled in humans (*Barban et al., 2016*; *Locke et al., 2015*). This lack of control means that some loci may be spuriously identified and others left undiscovered. More recent approaches, including transcriptome-wide association studies (TWAS) (*Wainberg et al., 2019*) and epigenome-wide association studies (EWAS) (*Rakyan et al., 2011*) recognize the intermediate role of epigenetic regulation and gene expression in driving complex traits and diseases. However, TWAS approaches focus specifically on predicted genetic effects on expression to prioritize GWAS candidates and thus do not account for potentially causal environmentally induced changes in gene expression. EWAS approaches, like GWAS and TWAS, are performed in individuals for which genetic and environmental information is not experimentally controlled, and thus it is unclear if identified epigenomic changes are driven by a genetic difference or an environmental effect (*Battram et al., 2022*). Overall, the major approaches used to uncover the etiology of complex traits and diseases are limited in their ability to separate genetic and environmental effects. Here, we show that even stochastic differences among individuals can be strongly predictive of complex reproductive traits when analyzed through the lens of differences in gene expression.

Complete control of genetic background, especially as layered across different environment inputs, is impossible within human populations. Model organism studies have therefore been invaluable in building our understanding of environmentally induced effects on epigenetic regulation, gene expression, and traits of interest in a controlled genetic background (*Torres-Garcia et al., 2020*; *Lismer et al., 2021*; *Kaletsky et al., 2020*; *DiVito Evans et al., 2023*; *Werner et al., 2023*; *Öst et al., 2014*; *Webster and Phillips, 2025*). However, such analyses generally focus on bulk populations (typically either to ensure ample material or to average differences across individuals within a replicate for model organisms such as *Drosophila* and *Caenorhabditis elegans*; *Meyer and Schumacher, 2021*; *Daines et al., 2011*), or they use very few individuals (typically due to difficulty sampling many larger organisms such as mice; *Li et al., 2017*), precluding knowledge of how individual life trajectories may influence gene expression, epigenetic information, and traits of interest. In recent years, it has become possible to generate individual transcriptomic profiles in *C. elegans* and *Drosophila*, and methods to profile individuals in high throughput have improved (*Chang et al., 2021*; *Serra et al., 2018*; *Pallares et al., 2020*; *Lin et al., 2016*; *Werkhoven et al., 2021*; *Wang et al., 2022*; *Perez et al., 2017*). This has revealed that genetically identical individuals in a shared environment indeed differ in their gene expression profiles. However, the effects that gene expression differences across genetically identical individuals may have on complex fitness-related phenotypes of these same individuals – which also vary substantially across genetically identical individuals – remain largely unknown. To fully separate genetic and environmental causation, we use an isogenic population of *C. elegans* to show that individuals in a common environment at the same developmental stage exhibit substantial differences in two fitness-related reproductive traits. By performing mRNA-seq on each individual worm, we identify hundreds of genes for which variation in expression is strongly associated with reproductive trait variation, determine the extent to which expression variation is driven by noise versus known environmental history, identify small sets of genes that are highly predictive of both reproductive traits, and show experimentally that reducing mRNA levels of individual genes causally affects progeny production. Finally, we show that predictive gene sets for reproductive traits are enriched for genes with the chromatin modification H3K27me3, revealing that chromatin environment defines which loci are subject to the individual variation in gene expression that drives variation in complex traits.

## Results

To determine if differences in gene expression levels correspond to differences in complex traits, we measured reproductive traits and whole-animal mRNA transcriptomes for each of 180 isogenic *C. elegans* in a shared environment at the same stage of early adulthood (*Figure 1*). Two reproductive traits – egg-laying onset (ELO) and the number of progeny produced in the first 24 hr of egg-laying (early brood) – were measured for each individual. To generate additional phenotypic variation and allow us to distinguish between gene expression noise and known environmental sources of variation, worms experienced controlled environmental perturbations early in development in a two-factor design: they were either born to a day 1 or 3 adult parent (corresponding to the beginning and middle of egg-laying in *C. elegans*, respectively), and, once hatched, they were subject to either a

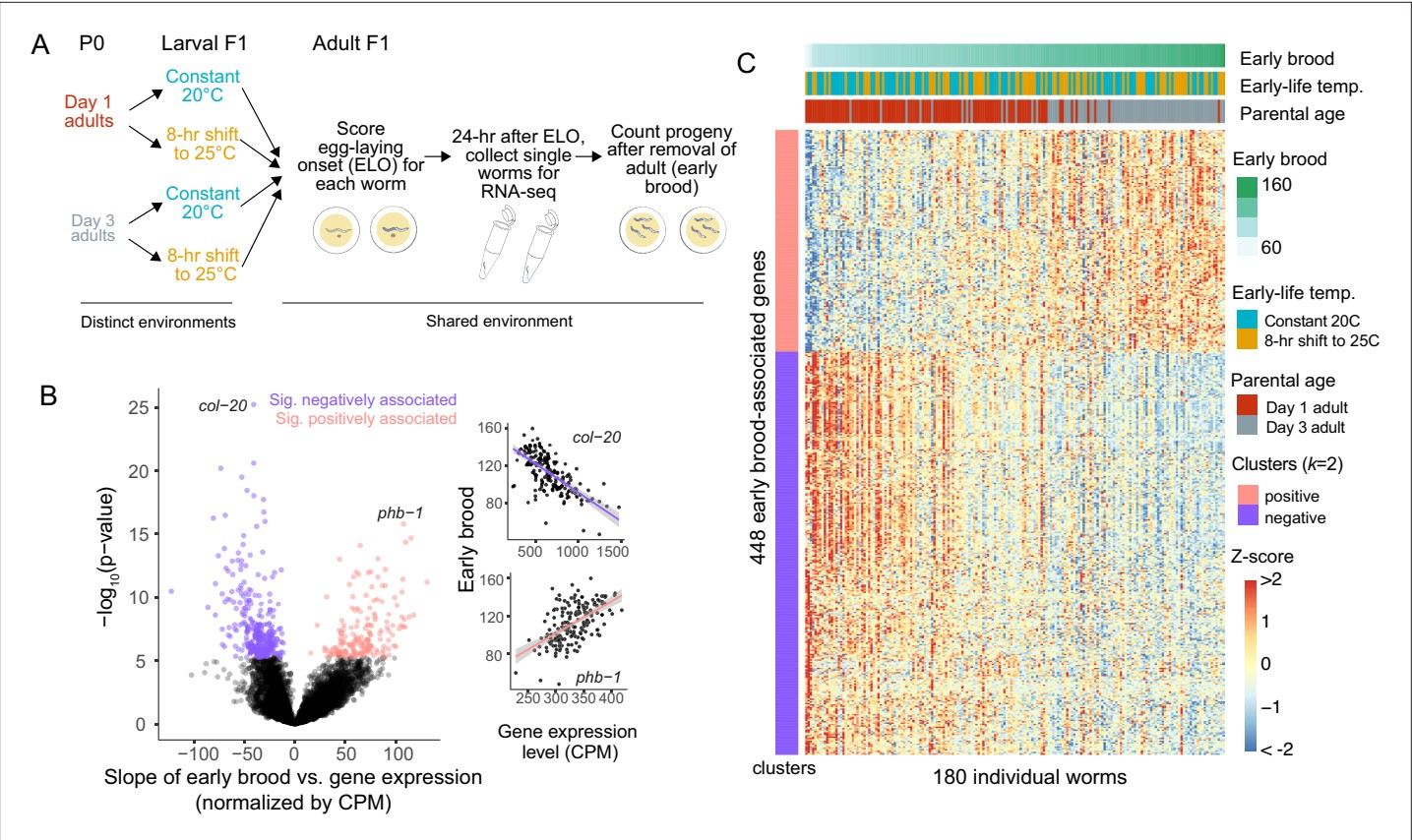

**Figure 1.** Differences in early brood across isogenic individuals are associated with expression levels of hundreds of genes. (**A**) Experimental design for single-individual mRNA-seq. P0 worms are allowed to lay progeny as day 1 or 3 adults. F1 progeny are either subject to a constant temperature or an 8-hr shift to 25°C. Beginning in mid-larval stages, all F1 worms experience the same environment. For each F1 worm, time to egg-laying onset is scored. 24 hr after egg-laying onset, each F1 worm is collected for single-individual mRNA-seq. F2 progeny laid on plates in the first 24 hr of egg-laying are counted to determine the early brood of each F1 worm. (**B**) Each of 8824 expressed genes was analyzed in a mixed model to assess the strength of its association with early brood. Significant genes are in pink (positively associated) or purple (negatively associated), non-significant genes are in black. Significance determined by a Bonferroni-corrected p-value cutoff of 0.05 (nominal p-value of $5.67 \times 10^{-6}$). Phenotypic and expression data for each worm is shown for the top two genes, *col-20* and *phb-1*, with linear model fits in black and 95% confidence intervals in gray. (**C**) Heatmap of 448 brood-associated genes for all 180 individuals. Worms are sorted from left to right from lowest to highest early brood. Early brood data and environmental information is shown in the bars at the top. Genes are sorted into two clusters based on whether the association was positive or negative.

The online version of this article includes the following figure supplement(s) for figure 1:

**Figure supplement 1.** Isogenic worms at the same developmental stage exhibit variability in egg-laying onset and early brood that partially depends on previous environmental conditions.

**Figure supplement 2.** Variation in 11 genes is significantly associated with egg-laying onset.

constant 20°C temperature or an 8-hr shift to 25°C (both corresponding to typical rearing temperatures for *C. elegans*). This design is ideal for at least two reasons. First, in standard worm rearing, worms that are at the same developmental stage in a shared environment have experienced some differences in their environmental histories that are typically unknown to the observer because they appear physiologically identical. Some of these differences in environmental history include factors that we directly controlled in this experiment, including the age of each worm's parent and small fluctuations in rearing temperature. By explicitly incorporating this *historical* environmental heterogeneity, we can distinguish between gene expression variability induced by environmental history and gene expression variability driven by 'noise'. Noise encompasses both unknown extrinsic factors (e.g., microenvironmental differences in food availability) and intrinsic factors (e.g., stochasticity in biochemical reactions). Second, with this approach, we can ask whether sequencing adult worms in a shared environment is a viable approach for phenotyping-by-transcriptomics and whether such an approach depends on knowledge of the historical environment. In other words, if an adult worm is picked off a

plate and sequenced, can we determine when that worm began reproducing and how many progeny it produced, without regard to its previous environment or its ancestors' environment? In our experiment, isogenic adult worms in a common environment (with distinct historical environments) exhibited a range of both ELO and early brood trait values (*Figure 1—figure supplement 1*), and variation in these traits was partially driven by environmental history, consistent with prior work on parental age and rearing temperature in *C. elegans* (*Perez et al., 2017*; *Gouvêa et al., 2015*; *Hirsh et al., 1976*; *Figure 1—figure supplement 1*). To determine if gene expression differences identified in single-worm mRNA-seq were directly associated with reproductive traits measured in the same worms, we used a linear mixed modeling approach and, remarkably, identified significant genes corresponding to each trait (*Figure 1B, C*, *Figure 1—figure supplement 2*). Expression of 448 genes was strongly associated with early brood (*Figure 1B, C*, Bonferroni p-value of 0.05, nominal p-value of $5.7 \times 10^{-6}$), while expression of 11 genes was strongly associated with ELO (*Figure 1—figure supplement 2*). This model was agnostic to the environmental history of the worm, but accounted for biological replicate, and we confirmed that these effects were not driven by mutations arising in the population over the course of the experiment (see Methods). This reveals many genes, particularly for early brood, for which expression differences across genetically identical individuals strongly associate with the reproductive traits of these individuals. Importantly, this indicates that expression levels of hundreds of genes provide useful information about the reproductive status of individual worms.

We next quantified the effect sizes on reproductive traits explained by the individual genes that were significant in the previous analysis. Further, because of our experimental design, we also determined

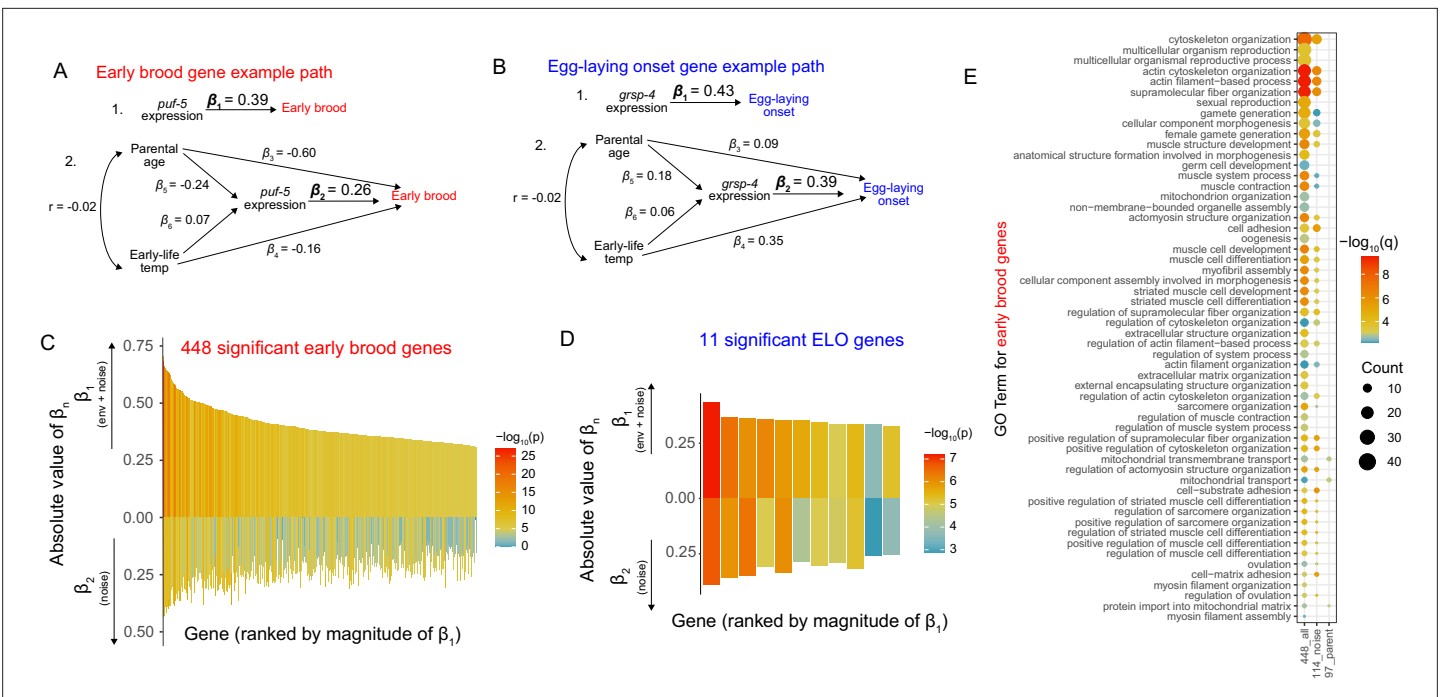

**Figure 2.** Associations between gene expression and reproductive traits are partially driven by both noise and historical environment. (**A**) Two example path analyses showing the relationship between gene expression and early brood. In Model 1, gene expression of a particular gene, *puf-5*, drives differences in early brood, and $\beta_1$ is the coefficient of the linear mixed model. $\beta_1$ represents the total effect of gene expression on early brood. In Model 2, $\beta_2$ represents the effect of *puf-5* on early brood that is independent of the historical environment. (**B**) Two example path analyses showing the relationship between gene expression and egg-laying onset, analogous to A. In Model 1, $\beta_1$ is the total effect of *grsp-4* on egg-laying onset, while in Model 2, $\beta_2$ represents the effect of *grsp-4* that is independent of historical environment. (**C**) The 448 genes for which expression variation is significantly associated with early brood, ordered by the magnitude of their $\beta_1$ effect sizes. The absolute values of $\beta_1$ and $\beta_2$ are plotted upward and downward, respectively, for each gene. (**D**) The 11 genes for which expression variation is significantly associated with egg-laying onset across individuals, ordered by the magnitude of their $\beta_1$ effect sizes. The absolute values of $\beta_1$ and $\beta_2$ are plotted upward and downward, respectively, for each gene. For C and D, legend indicates the $-\log_{10}$(p-value) of $\beta_1$ and $\beta_2$ for each gene. (**E**) Gene Ontology (GO) Terms for the 448 genes associated with early brood with a $q$-value <0.01 are shown. GO Terms for the two subsets of the 448 genes with overlapping GO Terms are also shown: 114 genes that have significant values of $\beta_2$ but parental age does not have a significant effect on expression ('noise' genes), and 97 genes that do not have significant values of $\beta_2$ but parental age does have a significant effect on expression (parental age genes).

the effect size and significance of the association after accounting for environmental history in our model, allowing us to distinguish how much of the effect of gene expression on reproductive traits is gene expression 'noise' and not driven by known environmental history. At the phenotypic level, differences in early brood were partially driven by parental age but not affected by early-life temperature (*Figure 1C*, *Figure 1—figure supplement 1*). Differences in the timing of ELO were partially driven by both parental age and temperature. Because temperature and parental age have effects on progeny phenotypes, we wanted to test the extent to which the association between gene expression and reproductive traits was driven by environmental history (parental age or early-life temperature) for the identified genes. Previous work on parental age found that maternal age affects vitellogenin gene expression to impact progeny phenotypes (*Perez et al., 2017*), but how other genes may be affected is not yet known. To determine effect sizes, we calculated the standardized beta coefficient for each gene using a linear mixed model without environmental history, in line with the analysis done in *Figure 1* ($\beta_1$ in *Figure 2A, B*). We then generated a linear mixed model that incorporates environmental history and calculated the standardized beta for the expression component of the model, effectively isolating gene expression 'noise' that is not explained by environmental history ($\beta_2$ in *Figure 2A, B*). These two approaches to calculating standardized beta values for expression are illustrated as path analyses in *Figure 2A, B*. Model 1 considers expression alone and its association with brood, while Model 2 is a causal model that incorporates environmental and its effects on gene expression. Equations for these models, including the coefficients for $\beta_3$–$\beta_6$, can be found in the methods. Values for all coefficients can be found in . Here, we primarily focus on standardized beta values $\beta_1$ and $\beta_2$, because these, respectively, represent (1) the total effect of expression on each trait and (2) the effect of expression on each trait after controlling for historical environment. In *Figure 2A*, *puf-5* is shown as an example gene that is part of the group of 448 genes significantly associated with brood. The $\beta_1$ value of 0.39 for *puf-5* is significant and indicates that a 1 standard deviation change in gene expression is associated with a 0.39 standard deviation change in early brood. The $\beta_2$ value of 0.26 for *puf-5* is also significant, meaning that after accounting for historical environment, a 1 standard deviation change in gene expression is associated with a 0.26 standard deviation change in early brood. The standard deviation in early brood is 20.3, so this indicates that these gene expression changes associated with *puf-5* account for approximately five to eight progeny produced in the first day of egg-laying, which is a sizable effect on fitness. Similarly, in *Figure 2B*, *grsp-4* is shown as an example gene with a significant association with ELO both before and after accounting for environmental history. The absolute values of $\beta_1$ and $\beta_2$ for all significant genes are shown in *Figure 2C, D*. Early brood genes had a median absolute value of $\beta_1$ of 0.38 and a median absolute value of $\beta_2$ of 0.23, indicating that for a 1 standard deviation change in gene expression for a typical significant gene, there was a corresponding 0.38 or 0.23 standard deviation change, respectively, in early brood. ELO genes had a median $\beta_1$ value of 0.35 and $\beta_2$ value of 0.31. Thus, while we are able to identify many more genes for early brood compared to ELO, genes for both traits have largely comparable effect sizes. Among significant genes for both traits, $\beta_2$ values were consistently lower than $\beta_1$ (*Figure 2C, D*), suggesting some of the total effect size was driven by environmental history rather than pure noise. Nonetheless, $\beta_2$ values were often still highly significant with substantial explanatory power, indicating that gene expression noise does indeed explain differences in reproductive traits. For ELO, 5 of 11 genes have highly significant values of $\beta_2$ (Bonferroni-corrected p-value of less than 0.05), though all are nominally significant. Of the 448 genes associated with early brood, 157 genes have highly significant values of $\beta_2$, indicating that gene expression noise independent of historical environment is driving associations for these genes.

In our previous analysis, we identified genes for which variation is associated with early brood or ELO and determined which of these genes had expression variation that was not explained by the historical environment (i.e., the 157 noise genes). Because of the unique structure of our dataset, we can also directly ask whether the historical environment has effects on gene expression, regardless of whether these expression changes lead to changes in our reproductive traits. To address this, we used a negative binomial mixed model to identify genes that are differentially expressed due to parental age or early-life temperature. We identified 186 genes with expression changes significantly affected by parental age, and just 2 genes significantly affected by early-life temperature. 140 of the 186 genes differentially expressed due to parental age in the previous generation overlapped with the original 448 genes for which expression variation is associated with early brood, suggesting that the historical

environment, that is, parental age, at least partially accounts for the gene expression changes that are associated with early brood. Forty-three of the 186 genes also overlap with our 'noise' genes, meaning they exhibit gene expression variability significantly explained by *both* historical environment and by noise; that is, these two sources of variation operate together for these genes. Thus, among genes for which expression variation is associated with early brood, we can gain insight on whether this variability is primarily driven by expression 'noise', historical environmental perturbations, or both.

We next assessed Gene Ontology (GO) enrichments for genes associated with reproductive trait variation. We focused on early brood genes because hundreds of genes were identified, giving us power to detect enriched terms. We evaluated GO Terms for all 448 genes as well as two subsets of the 448 genes: (1) genes that have a significant value of $\beta_2$ and a non-significant effect of parental age on gene expression (114 genes, the 'noise' group) and (2) genes that have a non-significant value of $\beta_2$ and a significant value of parental age on gene expression (97 genes, the 'parental age' group). Enrichments for all three gene groups are shown in *Figure 2E*. We find that the majority of enriched terms for the 448 genes are also enriched in either the noise group or parental age group, in a mutually exclusive manner (i.e., no term is enriched in both subsets), suggesting that these two sources of gene expression variation may, to some extent, be regulated by distinct processes. Indeed, the enriched parental age terms all impinge on mitochondrial regulation, consistent with parental age affecting mitochondria-related gene expression in progeny. Notable categories among enriched noise terms include cytoskeleton organization and female gamete generation, with the latter category particularly relevant for affecting progeny production.

The analysis shown in *Figure 2* provides a framework to understand individual gene expression as useful both for prediction and understanding causal mechanisms. Toward the goal of using gene expression profiles to predict phenotypes, we next asked if we could use *aggregate* gene expression profiles for combinations of genes to predict the reproductive traits of individuals. For example, the $\beta_1$ value of a single gene provides an indication of to what extent the phenotype is associated with that single gene. If instead of generating individual regressions for each gene, we combined genes into a single model, would we be able to better predict phenotypes? Because we are focused on prediction rather than the distinction between expression noise and the historical environment in this analysis, we focus on an analysis that is agnostic to environmental history, but we emphasize that all worms are genetically identical and collected while in a shared environment at the same developmental stage. Principal component analysis (PCA) provides one such aggregate profile which simplifies gene expression data to linearly independent factors (principal components, PCs) that together explain the variation in the gene expression data and are ordered by how much variation they explain (i.e., PC1 explains the most variation, PC2 explains the second-most, etc.). We therefore first asked whether ordered PCs together in a multiple regression explain more variation than expected by chance. Indeed, gene expression PCs explained both ELO and early brood data substantially better than randomized ELO and early brood data, indicating that individual gene expression profiles provide a clear signal of phenotypic information (*Figure 3A*). Because we have phenotypic and transcriptomic data for 180 worms, and therefore up to 180 linearly independent PCs that can be added to a multiple regression model, the total $R^2$ eventually rises to 1 even for randomized data, as shown in *Figure 3A*. Therefore, the optimal number of PCs occurs at the inflection points of the graph, which is after only 7 PCs for early brood ($R^2$ of 0.55) but 28 PCs for ELO ($R^2$ of 0.56). Thus, for early brood in particular, very few PCs explain over half of the variation in the data and explain substantially more variation than for shuffled phenotypic data. Therefore, whole transcriptomes are a viable approach to trait prediction in isogenic *C. elegans*.

Because PCA incorporates data from all expressed genes, we next wanted to determine if a smaller subset of genes could be similarly predictive of reproductive traits. We again used a multiple regression approach in which we successively selected each gene by identifying the gene that explained the most additional variance compared to the previous model (*Figure 3B*). Genes that exhibit highly correlated expression patterns, even if they are each individually associated with traits of interest, would not explain much additional variance in a multiple regression approach. Therefore, this approach prioritizes identifying genes that are most predictive in conjunction with other genes. We found relatively small sets of genes that predict ELO and early brood phenotypes very well and do so significantly better than randomized trait data, with diminishing returns for variance explained as the number of genes increases (*Figure 3B*). Maximum differences in the proportion of variance

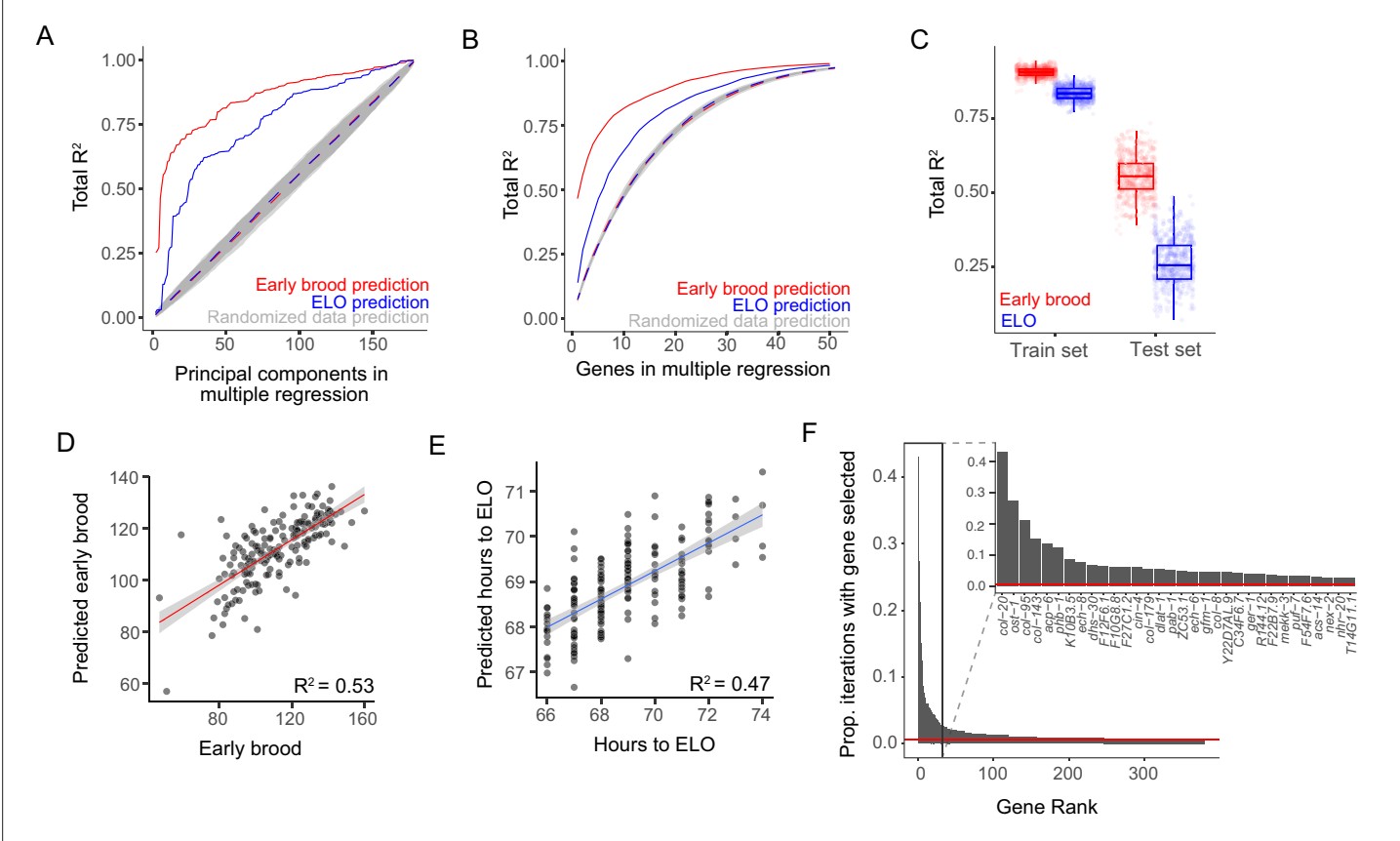

**Figure 3.** Multi-gene models are highly predictive of early brood and egg-laying onset (ELO) at an individual level. (**A**) Total $R^2$ of gene expression principal components (PCs) in a multiple regression. PCs explain early brood (red), ELO (blue), and randomized data (dotted lines and gray shading). (**B**) Total $R^2$ of top genes in a multiple regression. Gene sets explain early brood (red), ELO (blue), and randomized data (dotted lines and gray shading). For A and B, dotted lines represent average of randomized data iterations and gray shading indicates standard deviation. (**C**) Total $R^2$ of top 10 genes in a multiple regression identified in a train set and total $R^2$ of the same 10 genes used in a test set. Train and test sets randomly selected 500 times and top 10 genes identified in each iteration. Box plot center lines indicate median and box limits indicate upper and lower quartiles. An example of one iteration of train and test sets and corresponding model fits is shown for early brood. For D and E, machine learning model elastic net regression and leave-one-out cross-validation were used to identify a predicted trait for each worm given transcriptomic profile and compared to true trait data. Linear fit used to determine $R^2$. Lines indicate linear fit, and gray indicates 95% confidence intervals. (**D**) Early brood model. (**E**) ELO model. (**F**) Proportion of 500 iterations from C in which a given gene is selected as one of the top 10 predictive genes for early brood, ordered by those selected from most to least often, and an inset showing the genes that are most frequently selected as predictive.

explained between trait and randomized data occur in a model with only four genes for the early brood analysis ($R^2$ difference of 0.433) and with seven genes in the ELO analysis ($R^2$ difference of 0.195). While less powerful than utilizing the entire transcriptome, this result highlights that individual worm gene expression profiles consisting of only a handful of key selected genes are highly informative for predicting the individual's traits.

We next wanted to develop a quantitative estimate of how much trait variation is explained by aggregate gene expression profiles using two distinct approaches that prevent model overfitting and would therefore be more feasible to use for trait prediction in a new dataset. First, as an extension of the multiple regression analysis identifying predictive genes, we randomly split the full dataset (consisting of trait data and gene expression profiles) into training and test sets. In the training set, we used the multiple regression approach to identify the set of 10 genes that explained the most variance in the trait data. We then asked how well this set of 10 genes explained variance in the test set and repeated this process 500 times, each time randomly splitting the data into training and test sets (*Figure 3C*). We found that the median proportion of variance explained in the test sets was 0.55 (range 0.36–0.73) for early brood and 0.26 (range 0.07–0.49) for ELO, and an example of the fit of one of these iterations is also shown (*Figure 3C*). Thus, sets of only 10 genes explain over half of

trait variation in early brood and over a quarter of variation in ELO using test sets of data that were not used to select predictive genes. All 500 sets of 10 predictive genes are available as part of . As a second approach, we used elastic net regression and leave-one-out cross-validation on the full set of expressed genes. In other words, we used a machine learning approach designed to prevent overfitting to develop a model for all individuals except one, used the model to predict the trait for the remaining individual, then repeated for all individuals. Consistent with the multiple regression approach, the proportion of variance explained was 0.53 for early brood and 0.47 for ELO (*Figure 3D, E*). Aggregate gene expression profiles identified via multiple approaches are thus highly predictive of reproductive traits among isogenic individuals in a shared environment, reliably explaining about half of reproductive trait variation among genetically identical individuals.

A key remaining question is whether predictive genes are causal to affect phenotypes and whether similar underlying regulation underlies expression variation in predictive genes. To address causality, we focused on the early brood trait because predictive genes explained a higher proportion of variance for this trait compared to ELO. We used RNA interference (RNAi) to reduce the level of gene expression of predictive genes and prioritized genes that were also in the 448 brood-associated genes, either highly independent of parental age or dependent on parental age, and have an ortholog in humans. Within this curated set, genes causally affected early brood in five of seven cases compared to empty vector (*Figure 4*). We followed up on one pair of genes from this screen, *puf-5* and *puf-7*, to ask whether perturbing the dose of RNAi shows a dose response effect on brood. These genes have previously been shown to act redundantly in fertility in *C. elegans* (*Stumpf et al., 2008*; *Lublin and Evans, 2007*) and when knocked down simultaneously in our assay have a large effect on brood without causing sterility (*Figure 4A*). Consistent with the idea that differences in the level of mRNA affect the early brood trait, the dose of RNAi corresponds to the effect size on brood (*Figure 4B*). Thus, this analysis reveals that the predictive genes also make excellent candidates as causal effectors of complex traits and can affect these traits dependent on the amount of mRNA available.

Expression of predictive genes can be causally related to reproductive traits (*Figure 4A, B*), but do these genes share underlying regulation? The majority of *C. elegans* genes fall into one of two chromatin domains with distinct levels of gene activity (*Jänes et al., 2018*; *Ahringer and Gasser, 2018*; *Evans et al., 2016*). Regulated domains include genes that are expressed at specific stages of development and are marked by the facultative heterochromatin modification H3K27me3. In contrast, active domains include genes that are expressed broadly across cell types, including in the germline, and marked by the euchromatic modification H3K36me3 (*Jänes et al., 2018*; *Evans et al., 2016*; *Serizay et al., 2020*). These two marks are generally mutually exclusive, and marks are robust across several stages of development in *C. elegans* (*Evans et al., 2016*) despite major changes in gene expression that occur during this time. Importantly, the robustness of these chromatin marks throughout development means that we can use existing data to determine whether genes in active or regulated domains are differentially associated with gene expression variation linked to reproductive traits in our study. We therefore used genes previously classified as regulated or active to gain insights into the epigenetic status of genes associated with and predictive of reproductive traits. Because active genes are enriched for being expressed in the germline, we performed a two-factor analysis that takes into account both tissue (soma or germline) and chromatin domain (regulated and active). Among early brood-associated genes, germline genes are strongly positively associated with brood, regardless of chromatin environment, but chromatin environment and tissue also exhibit a significant interaction (*Figure 4C*). Post hoc pairwise comparisons reveal that this interaction is driven by differences in the chromatin environment in the soma. In particular, while active somatic genes have on average no effect on brood, regulated somatic genes are strongly negatively associated. Because regulated genes are typically expressed in a temporally controlled or tissue-specific manner in the soma, this may indicate that aberrant expression of these genes has negative consequences for reproduction, or similarly, that appropriate regulation and silencing of these genes promotes reproduction. To determine if epigenetic regulation may underlie variation in predictive genes for both traits overall, without regard to the direction of association, we used all gene sets identified from iterations of predictive sets (*Figure 3C, F*) to determine whether these genes are more likely to be found in regulated domains. For both reproductive traits, predictive genes were strongly over-enriched for being in a regulated domain compared to randomly selected gene sets from the same background set (*Figure 4D*). Genes in regulated domains are more likely to be variable across developmental

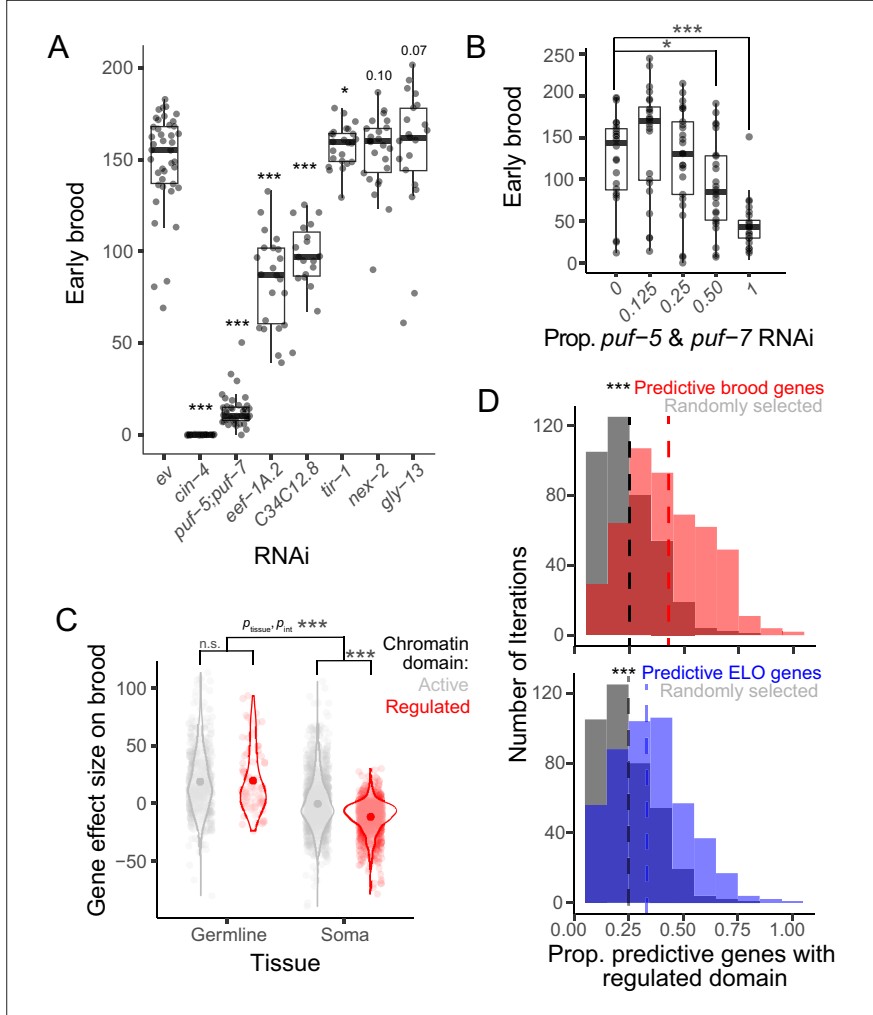

**Figure 4.** Genes predictive of reproductive traits causally affect early brood and are enriched for H3K27me3. (**A**) RNA interference (RNAi) knockdown of selected predictive genes compared to empty vector (ev) and effects on early brood. (**B**) Dose response of *puf-5* and *puf-7* RNAi together with empty vector and effects on early brood. For A and B, linear mixed model with RNAi as fixed effect and biological replicate as a random effect. Center lines, median; box limits, upper and lower quartiles, *p < 0.05, ***p < 0.001. (**C**) Effect sizes on brood for genes categorized by tissue (soma or germline) and chromatin domain (active or regulated). Two-way ANOVA showed significant interaction between tissue and chromatin domain (p = 1.58 × 10⁻⁶) and a significant main effect of tissue (p < 2 × 10⁻¹⁶). Post hoc Tukey tests corrected for multiple testing showed a significant difference between chromatin domains within somatic tissues (adjusted p = 0) but not germline tissues (adjusted p = 0.98). Within somatic genes, active somatic genes effects on average do not differ from 0 (p = 0.3, one-sample *t*-test with μ = 0), while regulated somatic genes on average have a significant negative association with brood (p < 2.2 × 10⁻¹⁶, one-sample *t*-test with μ = 0), ***p < 0.001. (**D**) Histogram of all iterations identifying predictive gene sets and the proportion of each set in a regulated chromatin domain marked with H3K27me3 compared to randomly selected sets of genes from the same background set. Early brood iterations are shown in pink, egg-laying onset (ELO) in blue, and the same gray control is shown for both histograms. The vertical lines are color-coded to indicate the median proportion of genes in regulated domains in the corresponding condition. Kolmogorov–Smirnov test used to determine statistical significance, ***p < 0.001.

The online version of this article includes the following figure supplement(s) for figure 4:

**Figure supplement 1.** Single-individual mRNA-seq reveals most and least variable genes after controlling for environmental perturbations and biological replicate.

stages and cell types (*Evans et al., 2016*), and we find in our data that genes with the most expression variation across individuals unexplained by experimental factors are also more likely to be found in regulated domains (*Figure 4—figure supplement 1*). Taken together, these results suggest that regulated domains are subject to gene expression variation in genetically identical individuals and that this variation is an important substrate underlying complex trait variation.

## Discussion

Individuals differ from each other for a variety of traits over the course of their lives, but being able to reliably predict these differences and understand their causal basis is a challenge. Gene expression is a critical intermediate functional layer between genotype and phenotype, but the role that gene expression plays in shaping individual organismal traits independently of genetic variation and environmental factors has remained elusive. Here, we effectively screened for genes that exhibit expression variation between isogenic *C. elegans* individuals and for which this variation corresponds to two critical reproductive traits – time to ELO and early brood. We identified hundreds of genes associated with these traits, differentiated between expression variation that is dependent and independent of historical environmental factors, and defined sets of highly predictive and causal genes. Predictive genes for both traits are enriched for a particular chromatin modification – H3K27me3 – highlighting the possibility that chromatin structure may underlie the potential for gene expression to vary between individuals. These findings build upon past research and suggest key areas for future work.

With our framework, we identify individual differences in gene expression that are associated with reproductive traits, and we parse whether these effects are primarily driven by noise or by known historical environmental factors. For example, *puf-5* and *puf-7* expression is highly associated with early brood after accounting for the known historical environment, and knockdown of these genes causally affects early brood in a dose-dependent manner. On the other hand, genes like C34C12.8 exhibit an overall association with early brood that is primarily driven by the effect of parental age on gene expression; nonetheless, C34C12.8 is also causally linked to early brood. Thus, by experimentally manipulating and measuring *individuals*, our approach is highly efficient in dissecting multiple sources of environmental variation – including noise and known environmental perturbations – underlying multiple traits that would otherwise be obscured by bulk experiments. In both cases, the mechanism(s) generating gene expression differences across individuals remains unknown. For 'noise' genes in particular, whether differences in gene expression are driven by extrinsic or intrinsic factors is unclear. While every effort was made to control the environment for our experiment, isogenic individuals nonetheless exhibit some extrinsic differences in their lives, such as how they choose to wander on a plate of food, that may to some extent impact gene expression and reproduction. We further expect that some of the variation observed is intrinsic, based on literature exploring the stochastic nature of transcription (*Swain et al., 2002*), as well as examples that suggest the amount of phenotypic variation present among individuals of a given genetic background is itself genetically controlled (*Yang et al., 2022*). Continued experimentation to determine the extent to which intrinsic or extrinsic factors regulate gene expression variability in this context – where such variability is directly related to complex phenotypes – is of high interest for future work.

When gene expression differences are identified across individuals by single-worm mRNA-seq, such differences could occur because individuals have different numbers of cells of a given type (e.g., germ cells) but each cell expresses genes at consistent levels, or alternatively, that individuals differ because within each cell, additional copies of a gene are being transcribed while cell numbers themselves remain constant. Of course, a combination of these scenarios is possible as well. *C. elegans* uniquely possess an invariant somatic cell lineage (*Sulston and Horvitz, 1977*), but can exhibit differences in cell number in the adult germline, including in the primordial zone (*Tolkin and Hubbard, 2021*) or in the uterus (*Mignerot et al., 2024*). Our results show that, in general, genes expressed in the germline tend to be positively associated with early brood, while somatic genes tend to be negatively associated with brood. Of somatic genes, genes marked with H3K27me3 are particularly predictive of early brood. Somatic genes marked with H3K27me3 are also more likely to exhibit gene expression variability in general. Given that the somatic lineage is invariant and somatic genes marked with H3K27me3 are the most variable in expression and most predictive of early brood, it is unlikely that cell number differences are the major driver of gene expression differences. While single-cell RNA-seq has been performed in *C. elegans*, such analysis requires thousands of individuals to be

pooled to obtain single cells and is not technically possible at the level of single individuals. Thus, to fully understand cell-type-specific variation in single *C. elegans*, and how these relate to individuals' traits, further innovation on this front is required.

We observed that differences in chromatin domains are associated with differences in gene expression and traits across individuals. Regulated genes are defined by the histone modification H3K27me3, a mark that is robust across developmental stages despite differences in gene expression, while active genes are marked with H3K36me3 and are typically broadly expressed, including in the germline. In our data, regulated genes are enriched both among genes with the most unexplained variation and among genes for which variation is predictive of reproductive traits. In contrast, germline genes that predominantly consist of genes in active domains are specifically positively associated with brood. Taken together, one possible model is that regulated genes marked with H3K27me3 are more subject to inter-individual variation than genes marked with H3K36me3, which may lead to differences in gene expression without changing the levels of these marks. This is consistent with observations that regulated genes exhibit a higher coefficient of variation across conditions and that increased expression of genes marked with H3K27me3 is not associated with a change in histone post-translational modifications (e.g., to an active mark like H3K36me3) (*Pérez-Lluch et al., 2015*). An alternative hypothesis is that the levels of the marks themselves vary and that this leads to changes in gene expression across individuals. While it is clear that regulated genes are more subject to inter-individual variability in expression, at this stage, it is unknown whether this variability is driven by differences in the level of chromatin marks. Performing analysis on histone modifications genome-wide in individual worms, which is still technically challenging, will be a key next step to determine whether differences in chromatin structure correlate with gene expression changes at an individual level.

Individual gene expression differences underlying reproductive traits suggest that gene expression differences may be important for a variety of other complex traits. Gene expression differences among isogenic individuals in a shared environment have been previously documented as predictive of lifespan (*Rea et al., 2005*; *Kinser et al., 2021*; *Mosley et al., 2025*). In these studies, reporter gene expression for specific candidate genes predicted individuals' later lifespan. Our results extend on such studies by identifying predictive genes in the context of reproductive traits in a high-throughput manner, rather than relying on specific reporters. Our approach can, in theory, be used directly in the context of other traits that occur throughout the life cycle, such as those related to development, but must be used indirectly for time-to-death traits like lifespan or stress resistance. Because a worm, once collected for RNA, cannot be subsequently phenotyped for traits like lifespan, using predictive reporters for time-to-death traits in conjunction with transcriptomics will likely be a useful approach for identifying novel regulators for a broad swath of traits.

Genetically, identical individuals can have large differences in phenotype if gene expression varies across those individuals. Our work illustrates that differences in gene expression can be causally linked to, and are highly predictive of, important reproductive phenotypes among genetically identical individuals. Assaying isogenic individuals for both organismal and molecular phenotypes enables us to uniquely understand the basis of non-genetic variation underlying individual differences and provides an avenue for high-throughput phenotyping-by-transcriptomics (*Webster and Phillips, 2025*). Given the major role of non-genetic factors in regulating complex traits and diseases in humans, identification of these molecular drivers of variation may be a powerful next step for advancing precision medicine.

## Methods

### Strains used

Strain N2-PD1073 was used for all experiments. N2-PD1073 is a subclone of the N2-derivative strain VC2010 that was generated in the process of assembling a new N2 reference. Worms were passaged on *E. coli* OP50 except during RNAi experiments, when worms were fed HT115 bacteria expressing dsRNA for genes of interest (see 'RNAi' section).

### Experimental setup for single-worm RNA-seq

Worms were well-fed at 20°C for several generations prior to beginning experiments. The experiment takes place over the course of 10 days. On day 1, for each of the five biological replicates, P0 embryos

and L4s were picked to separate 5 cm NGM plates with OP50 and kept at 20°C. Approximately 10–20 individuals were placed on each plate and several plates were picked for each stage, ensuring that worms did not experience dietary restriction. On day 4, P0 worms were either in the first day of adulthood (young adult) or third day of adulthood (older adult). P0 young adults and older adults were singled to new plates, allowed to lay progeny for 2 hr at 20°C, then removed from the plate. Hour 0 of development for F1 worms is considered the end of this egg-laying window. F1 worms were then incubated at 20°C until day 5 of the experiment (undergoing approximately 18 hr of development as L1/L2 larvae). On day 5, approximately half of these larvae remained at 20°C while the other half were shifted to 25°C for 8 hr then shifted back to 20°C. On day 6, all worms remained at 20°C and were singled to individual NGM plates with OP50 to ensure they can be individually phenotyped for reproductive traits. On day 7, from 66 to 74 hr of development, ELO was assessed every hour by checking for embryos on each plate. Worms that died, bagged, or otherwise did not reach ELO in this time interval were not included in subsequent analysis, and altogether this affected 6% of worms across all biological replicates. Time to ELO was noted for each worm. 24 hr after ELO, on day 8, each worm was collected for cell lysate generation for mRNA-seq (see 'Cell lysate generation' section of Materials and methods). On day 10, the number of progeny laid on each plate during the 24 hr between ELO and collection was counted as late larvae/young adults to ensure all progeny were counted. All together, each F1 worm collected experienced two environmental perturbations (young adult or older adult parent, and early-life constant temperature or temperature shift) and was measured for two traits (ELO time and number of progeny produced within 24 hr) prior to mRNA-seq.

## Cell lysate generation

Individual worms were collected for chemical lysis 24 hr after ELO (i.e., worms that reached ELO at 72 hr were collected at 96 hr, worms reaching ELO at 73 hr were collected at 97 hr, etc. for each hour). 200 µl of lysis mix, consisting of 187.2 µl Elution buffer (10 mM Tris, 0.1 mM EDTA) and 12.8 µl Proteinase K, was made fresh every 2–3 hr on ice. 6 µl of lysis mix was pipetted into each PCR tube according to how many worms needed to be collected in a given hour. Each worm was picked into a 10-µl drop of $dH_2O$ for a quick wash to remove any excess bacteria, then transferred to a second drop of $dH_2O$. After this, each worm was picked into an individual PCR tube with lysis mix. Immediately after worms are placed into tubes with lysis mix, we proceeded with the lysis protocol (typically no more than a dozen worms were prepared at once). The lysis protocol is 65°C for 10 min, 85°C for 1 min, and 4°C hold and is based on a previously published protocol (*Serra et al., 2018*). After the lysis protocol, worms were moved to ice and then stored at –80°C until mRNA-seq library preparation.

## Preparation and sequencing of mRNA-seq libraries

The KAPA mRNA-seq preparation kit (KK8580, Roche Sequencing Solutions) was used to prepare libraries. 5 µl of each cell lysate preparation described above, which contains total RNA, was used for each sample. Reagents were used at 0.25× and kit instructions were followed to generate libraries. Library concentrations were quantified using Qubit, and samples were pooled together in equimolar ratios in two batches of 96 for sequencing. Custom dual-indexed adapters were designed by the University of Oregon Genomics and Cell Characterization Core Facility, and sequencing data was generated with the Illumina NovaSeq 6000 at the same facility.

## Analysis of single-worm mRNA-seq data

FASTQ files were aligned to WBcel235 version of the *C. elegans* genome using subread-align (*Liao et al., 2013*) with flag -t 0. Bam files were counted using featureCounts with flag -T 5. Twelve of 192 samples were filtered out at this stage due to particularly low read counts (less than 1 million reads) and/or poor mapping quality (less than 70% of reads both mapped to the genome and assigned to a genomic feature). The remaining 180 samples had an average of 5.8 million reads (SD 3.4 million), and an average of 85.8% (SD 3.1%) of reads both mapping to the genome and assigned to a genomic feature. Resulting count data from these 180 samples was CPM-normalized in R using the package edgeR (*Robinson et al., 2010*) and only protein-coding genes were included. Depending on the downstream analysis, genes were filtered in one of two ways for further analysis: (1) genes expressed at a level of CPM >10 in at least one library (8824 genes) or (2) genes expressed at a level of CPM >1 across all libraries (7938 genes). While these two sets are largely overlapping, the

first way of filtering allows identification of genes that are not expressed at all in some samples but moderately expressed in others. However, as models become more complex, having some samples with 0 expression causes problems for model fitting and thus having a more stringently defined set is more appropriate. To identify genes associated with early brood and ELO, we looped through the set of 8824 genes and tested two linear mixed models using the 'nlme' package (**Pinheiro et al., 2024**) in R to test the effect of each gene on each of two reproductive phenotypes. In both cases, the fixed effect was the normalized gene expression level in each worm and the random effect was biological replicate. For the first model, the dependent variable was early brood, and for the second model, the dependent variable was hours to ELO. To determine the effect of environmental perturbations on gene expression, we looped through each gene and fit a negative binomial mixed model to each gene. In this case, for each gene, fixed effects included environmental perturbations (parental age and early-life temperature), the random effect included biological replicate, and the dependent variable was gene expression level. For both sets of models, the summary() function was used to generate p-values, and a Bonferroni correction was used to determine genes with significant effects. Gene expression variation that was unexplained by the negative binomial (residual variance compared to the model) was used to determine the most and least variable genes in the data after controlling for environmental effects and biological replicate. Because variance increases with higher expression, we identified most and least variable genes by fitting a loess model to $\log_{10}$(unexplained variance) versus $\log_{10}$(mean CPM) for the 7938 genes. Z-scores were calculated for each gene relative to the loess fit to determine genes that were most and least variable given their level of gene expression.

## Path analysis and effect size calculations

Using the R package 'lme4' (**Bates et al., 2015**), for each individual $i$ and biological replicate $j$, the following linear mixed models were fit for each gene $k$:

(Early brood)$_{ij}$ = $\beta_0$ + $\beta_1$(Expression level of gene $k$) + $u_j$ + $e_{ij}$

(Early brood)$_{ij}$ = $\beta_0$ + $\beta_2$(Expression level of gene $k$) + $\beta_3$(Parental age) + $\beta_4$(Early-life temperature) + $u_j$ + $e_{ij}$

(Egg-laying onset)$_{ij}$ = $\beta_0$ + $\beta_1$(Expression level of gene $k$) + $u_j$ + $e_{ij}$

(Egg-laying onset)$_{ij}$ = $\beta_0$ + $\beta_2$(Expression level of gene $k$) + $\beta_3$(Parental age) + $\beta_4$(Early-life temperature) + $u_j$ + $e_{ij}$

(Expression level of gene $k$) = $\beta_0$ + $\beta_5$(Parental age) + $\beta_6$(Early-life temperature) + $u_j$ + $e_{ij}$

Parental age was binarized such that progeny of day 1 adults were given the value of 1 and progeny of day 3 adults were given the value of 0. Early-life temperature was binarized such that individuals experiencing a constant 20°C temperature throughout their lives were given the value of 1 and individuals that experienced a 25°C temperature shift were given the value of 0. All values of $\beta_n$ were standardized such that a 1 standard deviation change in the independent variable represents a shift in the dependent variable corresponding to the value of $\beta_n$. $u_j$ represents the random intercept for each biological replicate, and $e_{ij}$ is the error term. Coefficients $\beta_1$ through $\beta_6$ are shown in a path analysis format in **Figure 2A, B**, and coefficients $\beta_1$ and $\beta_2$ are shown graphically in **Figure 2C, D** for all significant genes from **Figure 1**, **Figure 1—figure supplement 2**. The values of all coefficients for all significant genes are in.

## GO analysis

GO Term analysis was performed using the R package 'clusterProfiler' (**Yu et al., 2012**). The enrichGO function was used for each gene list with the following parameters: OrgDb = org.Ce.eg.db, keyType = "SYMBOL", ont = "BP", pAdjustMethod = "BH", qvalueCutoff = 0.05. Output for all three gene lists is included in . **Figure 2E** shows all enriched terms with a $q$-value <0.01 for the list of 448 genes and any enriched terms that overlap with this set from that are enriched from the subsets of 97 parental age genes or 114 noise genes. The set of 97 genes was generated by identifying genes in the set of 448 that were significantly affected by parental age and did not have a significant value of $\beta_2$. The set of 114 genes was generated by identifying genes in the set of 448 that were not significantly affected by parental age and had a significant value of $\beta_2$.

## Prediction analysis

PCA was performed on 7938 CPM-normalized genes (without regard to reproductive phenotype data). CPM values for each gene were mean-normalized and $\log_2$-transformed prior to use in the prcomp function in R. To determine whether PCs explained variance in the reproductive traits, each PC was successively added to a multiple linear regression model, beginning with the PC explaining the most variance in the gene expression data (i.e., PC1), to determine the total variance explained by cumulative PCs. As a control comparison, phenotypic data was shuffled repeatedly (100×) and the same multiple regression analysis was performed. This analysis was performed for both the early brood and ELO traits.

To identify a set of genes that explains variance in each trait, all 7938 genes were looped through in a linear regression model to identify the single gene explaining the most variance. Then the remaining 7937 genes were looped to identify which gene explained the most additional variance when the first gene was already included in the model. This was repeated until the total variance explained was close to 1. This analysis was repeated for shuffled phenotypic data as a control comparison.

To determine how well sets of genes would be likely to work in a new dataset, we used two approaches. First, we randomly split the data (including matched phenotypic and gene expression data) into two equal groups, the train set and test set, 500 times. Each time, for the train set, the protocol described above was used to identify the top 10 genes that together explain the most variance in the data using a multiple regression. This set of 10 genes was then used on the test set to determine how much variation is explained when the genes were not selected because of how well they work in this set. The typical variance explained in the test set provides an estimate of how well the genes should explain variance in a new dataset. In a second machine learning approach, we used elastic net regression using the glmnet package (*Friedman et al., 2010*; *Tay et al., 2023*) in R with alpha = 0.5 and leave-one-out cross-validation. That is, we trained the model for all data points except for one, then used the model to predict that remaining data point and repeated for all data points.

## Enrichment analysis

To determine whether brood-associated genes and variable gene sets were enriched for particular chromatin environments and tissue specificities, we used our previously generated annotation for *C. elegans* transcriptional units (*Wilson et al., 2023*). Briefly, in this approach, we used published data (*Jänes et al., 2018*; *Serizay et al., 2020*) to obtain tissue specificity and chromatin environment for *C. elegans* regulatory elements genome wide and then used bedtools2 (*Quinlan, 2014*) to annotate regulatory elements to the nearest gene using the coordinates of transcriptional units extracted from the Wormbase gff3 (version 279).

To determine if there was an interaction between tissue and chromatin environment for effects on early brood (*Figure 4C*), we restricted genes to those expressed in the soma or germline, and in either an active or regulated chromatin domain. We then performed a two-way ANOVA to determine if these factors affected the effect sizes for each gene on brood (the same effect sizes plotted in *Figure 1B*), followed by post doc Tukey's tests for pairwise comparisons and one-sample *t*-tests. To determine if gene expression variation differed by chromatin domain (*Figure 4—figure supplement 1*), we performed a Wilcoxon test comparing the variance *Z*-scores for regulated genes to the variance *Z*-scores for active genes.

To determine whether predictive genes were enriched for regulated or active genes as shown in *Figure 4D*, we used the 500 sets of 10 genes for each trait described in the 'Prediction analysis' section of the methods that were used to predict reproductive phenotypes (*Figure 3C*). We filtered the chromatin dataset described above to include only genes that were classified as either active or regulated. We then merged this with the background of 7938 expressed genes from our gene expression analysis. Of 10 predictive genes, a median of 7 genes was either active or regulated for each of the 500 iterations and included in subsequent analysis. For each iteration, the proportion of regulated genes was calculated. As a control, 500 random sets of 10 genes were selected from our background set, merged with the filtered chromatin data, and the proportion of regulated and active genes was determined. The distribution of the proportion of predictive regulated genes for each trait was compared to the proportion of regulated genes from randomly selected sets of genes using a Kolmogorov–Smirnov statistical test.

## RNAi

RNAi was performed by feeding. Colonies of RNAi bacteria (*E. coli* HT115 containing a specific RNAi construct) were individually grown in 5 ml LB with carbenicillin overnight (~16–18 hr) at 37°C with shaking. RNAi bacteria was obtained from the Ahringer library (*Kamath and Ahringer, 2003*) for empty vector (ev) and for genes of interest: ZK1127.7 (*cin-4*), F54C9.8 (*puf-5*), B0273.2 (*puf-6/7*), R03G5.1 (*eef-1A.2*), C34C12.8, F13B10.1 (*tir-1*), T07C4.9 (*nex-2*), and B0416.6 (*gly-13*). Plates used for RNAi were made with NGM, carbenicillin, and IPTG as described previously *Ahringer, 2006*. These plates were seeded with RNAi bacteria and allowed to dry and grow for ~30 hr. L4 larvae were plated in groups of ~10 on these plates, then singled the next day (~16–18 hr later) onto new RNAi plates prepared at the same time as the original batch of plates. Adult worms were allowed to lay progeny for 24 hr, then adults were removed from the plates. If a worm died during this time, it was censored. The number of progeny laid by each adult was counted 2 days later, resulting in an early brood measurement for these experiments. For the *puf-5* and *puf-7* dose response, *puf-5* and *puf-7* RNAi constructs were grown separately as described, then pooled together 1:1. This mixed RNAi was considered the full dose of *puf-5* and *puf-7* RNAi and was pooled with ev in the ratios shown in *Figure 4C* for the varying doses. Statistical analysis on early brood data was performed by comparing RNAi treatment targeting a particular gene to empty vector for biological replicates within a batch. Linear mixed effects models were used in the R package 'nlme' (*Pinheiro et al., 2024*) with RNAi treatment as a fixed effect and biological replicate as a random effect. The summary() function was used to generate p-values.

## Validation of isogenic worms

We used the mRNA-seq data to validate that the frequency of genetic variants among the essentially isogenic individual worms was very low. First, we used samtools (*Li et al., 2009*) to generate coverage files from the sorted bam files resulting from mRNA-seq data processing. We filtered coverage files for each worm to include only nucleotides that had at least 20 reads mapping to that nucleotide. For each worm and nucleotide, we determined whether the base call of that nucleotide was unambiguous (all 20+ reads have the same call), ambiguous with two genotypes (exactly two different bases were called), or ambiguous with more than two genotypes called. All worms with sufficient depth at a nucleotide were used to determine putative homozygous or heterozygous variants. A particular nucleotide was included in subsequent analysis if at least 10 worms had sufficient coverage at the site. This resulted in 8,361,705 unique sites and a total of 768,738,255 sites across all worms. Candidate homozygous variants included nucleotides that were called as one base unambiguously in at least one worm and called as a different base unambiguously in other worms. Candidate heterozygous variants were identified by filtering ambiguous nucleotides with two genotypes in which the frequency of each genotype was present at a frequency of between 0.3 and 0.7 and for which at least 10 other worms were unambiguously called a single genotype. These criteria reduce the possibility that a heterozygous mutant is erroneously called if a site was merely subject to rare sequencing errors. This resulted in a homozygous variant rate of $2.21 \times 10^{-8}$ and a heterozygous variant rate of $9.86 \times 10^{-7}$. While already quite low, these rates likely represent overestimates because visual inspection of bam files in some cases revealed ambiguous base calls. Over three quarters of sites (78.5%) with a putative heterozygous variant were found only in a single worm, meaning that these rare variants cannot account for an association with expression across 180 worms.

## Statistical analysis

Statistical tests are described in detail in figure legends and in the corresponding section of the methods. All individual data points are plotted in figures whenever possible.

## Acknowledgements

We thank Chien-Hui Chuang and the Bruce Bowerman lab for sharing RNAi clones. Funding was provided by grants F32GM146402 to AKW and R35GM131838 to PCP. We thank members of the Phillips lab for critical feedback. We would also like to thank WormBase.

## Additional information

### Funding

| Funder | Grant reference number | Author |
|---|---|---|
| National Institutes of Health | F32GM146402 | Amy K Webster |
| National Institutes of Health | R35GM131838 | Patrick C Phillips |

The funders had no role in study design, data collection, and interpretation, or the decision to submit the work for publication.

### Author contributions

Amy K Webster, Conceptualization, Data curation, Formal analysis, Funding acquisition, Investigation, Visualization, Writing – original draft; John H Willis, Erik Johnson, Investigation; Peter Sarkies, Formal analysis, Writing – review and editing; Patrick C Phillips, Conceptualization, Formal analysis, Supervision, Funding acquisition, Methodology, Project administration, Writing – review and editing

### Author ORCIDs

Amy K Webster ⓘ https://orcid.org/0000-0003-4302-8102
John H Willis ⓘ https://orcid.org/0000-0002-4263-7347
Peter Sarkies ⓘ https://orcid.org/0000-0003-0279-6199
Patrick C Phillips ⓘ https://orcid.org/0000-0001-7271-342X

Reviewer #1 (Public review): https://doi.org/10.7554/eLife.106525.3.sa1
Reviewer #2 (Public review): https://doi.org/10.7554/eLife.106525.3.sa2
Reviewer #3 (Public review): https://doi.org/10.7554/eLife.106525.3.sa3
Author response https://doi.org/10.7554/eLife.106525.3.sa4

## Additional files

### Supplementary files

MDAR checklist

Source data 1. Excel file containing data used to generate all figures.

### Data availability

All raw mRNA-seq data is available at NCBI GEO accession GSE244875. All data used to generate figures are available in *Source data 1*. Scripts are available at GitHub and have been preserved on Zenodo at time of publication.

The following dataset was generated:

| Author(s) | Year | Dataset title | Dataset URL | Database and Identifier |
|---|---|---|---|---|
| Webster AK, Willis JH, Johnson E, Phillips PC | 2025 | mRNA-seq of individual genetically identical *Caenorhabditis elegans* adults | http://www.ncbi.nlm.nih.gov/geo/query/acc.cgi?acc=GSE244875 | NCBI Gene Expression Omnibus, GSE244875 |

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

# Appendix 1

**Appendix 1—key resources table**

| Reagent type (species) or resource | Designation | Source or reference | Identifiers | Additional information |
|---|---|---|---|---|
| Strain, strain background (*Caenorhabditis elegans*) | N2-PD1073 | https://doi.org/10.17912/micropub.biology.000518 | | |
| Strain, strain background (*E. coli*) | HT115-L4440 | Caenorhabditis Genetics Center | | Empty vector |
| Strain, strain background (*E. coli*) | HT115-puf-5 (F54C9.8) | Ahringer RNAi library | | |
| Strain, strain background (*E. coli*) | HT115-puf-6/7 (F18A11.1, B0273.2) | Ahringer RNAi library | | |
| Strain, strain background (*E. coli*) | HT115-cin-4 (ZK1127.7) | Ahringer RNAi library | | |
| Strain, strain background (*E. coli*) | HT115-eef-1A.2 (R03G5.1) | Ahringer RNAi library | | |
| Strain, strain background (*E. coli*) | HT115-C34C12.8 | Ahringer RNAi library | | |
| Strain, strain background (*E. coli*) | HT115-tir-1 (F13B10.1) | Ahringer RNAi library | | |
| Strain, strain background (*E. coli*) | HT115-nex-2 (T07C4.9) | Ahringer RNAi library | | |
| Strain, strain background (*E. coli*) | HT115-gly-13 (B0416.6) | Ahringer RNAi library | | |
| Strain, strain background (*E. coli*) | OP50 | Caenorhabditis Genetics Center | | |
| Commercial assay or kit | KAPA mRNA HyperPrep Kit | Roche Sequencing Solutions | KK8580 | |
| Software, algorithm | Subread | https://subread.sourceforge.net/ | RRID:SCR_009803 | |
| Software, algorithm | edgeR | https://doi.org/10.18129/B9.bioc.edgeR | RRID:SCR_012802 | |
| Software, algorithm | nlme | https://cran.r-project.org/package=nlme | RRID:SCR_015655 | |
| Software, algorithm | lme4 | https://cran.r-project.org/package=lme4 | RRID:SCR_015654 | |
| Software, algorithm | clusterProfiler | https://doi.org/10.18129/B9.bioc.clusterProfiler | RRID:SCR_016884 | |
| Software, algorithm | bedtools2 | https://bedtools.readthedocs.io/en/latest/index.html | RRID:SCR_006646 | |
| Software, algorithm | samtools | https://www.htslib.org/ | RRID:SCR_002105 | |
| Software, algorithm | glmnet | https://cran.r-project.org/package=glmnet | RRID:SCR_015505 | |

