## [Editor Report · eLife Assessment]

This **important** study addresses the role of non-genetic factors in individual differences in phenotype. Using *C. elegans*, the study finds that non-genetic differences in gene expression, partly influenced by the environment, correlate with individual differences in two reproductive traits. This supports the use of gene expression data as a key intermediate for understanding complex traits. The clever study design makes for **compelling** evidence.

---

## [Referee Report · Reviewer #1 (Public review)]

Summary:

Genome-wide association studies have been an important approach to identifying the genetic basis of human traits and diseases. Despite their successes, for many traits, a substantial amount of variation cannot be explained by genetic factors, indicating that environmental variation and individual 'noise' (stochastic differences as well as unaccounted for environmental variation) also play important roles. The authors' goal was to address how gene expression variation in genetically identical individuals, driven by historical environmental differences and 'noise', could be used to predict reproductive trait differences.

Strengths:

To address this question, the authors took advantage of genetically identical *C. elegans* individuals to transcriptionally profile 180 adult hermaphrodite individuals that were also measured for two reproductive traits. A major strength of the paper is in its experimental design. While experimenters aim to control the environment that each worm experiences, it is known that there are small differences even when worms are grown together on the same agar plate - e.g., the age of their mother, their temperature, the amount of food they eat, and the oxygen and carbon dioxide levels depending on where they roam on the plate. Instead of neglecting this unknown variation, the authors design the experiment up front to create two differences in the historical environment experienced by each worm: (1) the age of its mother and (2) 8 8-hour temperature difference, either 20 or 25 C. This helped the authors interpret the gene expression differences and trait expression differences that they observed.

Using two statistical models, the authors measured the association of gene expression for 8824 genes with the two reproductive traits, considering both the level of expression and the historical environment experienced by each worm. Their data supports several conclusions. They convincingly show that gene expression differences are useful for predicting reproductive trait differences, predicting ~25-50% of the trait differences depending on the trait. Using RNAi, they also show that the genes they identify play a causal role in trait differences. Finally, they demonstrate an association with trait variation and the H3K27 trimethylation mark, suggesting that chromatin structure can be an important causal determinant of gene expression and trait variation.

Overall, this work supports the use of gene expression data as an important intermediate for understanding complex traits. This approach is also useful as a starting point for other labs in studying their trait of interest.

Weaknesses:

There are no major weaknesses that I have noted. Some important limitations of their work are worth highlighting, though (and I believe the authors would agree with these points):

(1) A large remaining question in the field of complex traits remains in splitting the role of non-genetic factors between environmental variation and stochastic noise. It is still an open question which role each of these factors plays in controlling the gene expression differences they measured between the individual worms.

(2) The ability of the authors to use gene expression to predict trait variation was strikingly different between the two traits they measured. For the early brood trait, 448 genes were statistically linked to the trait difference, while for egg-laying onset, only 11 genes were found. Similarly, the total R2 in the test set was ~50% vs. 25%. It is unclear why the differences occur, but this somewhat limits the generalizability of this approach to other traits.

(3) For technical reasons, this approach was limited to whole worm transcription. The role of tissue and cell-type expression differences is important to the field, so this limitation is relevant.

Comments on revisions: The authors have addressed my previous comments to my satisfaction.

---

## [Referee Report · Reviewer #2 (Public review)]

This paper measures associations between RNA transcript levels and important reproductive traits in the model organism *C. elegans*. The authors go beyond determining which gene expression differences underlie reproductive traits, but also (1) build a model that predicts these traits based on gene expression and (2) perform experiments to confirm that some transcript levels indeed affect reproductive traits. The clever study design allows the authors to determine which transcript levels impact reproductive traits, and also which transcriptional differences are driven by stochastic vs environmental differences. In sum, this is a comprehensive study that highlights the power of gene expression as a driver of phenotype, and also teases apart the various factors that affect the expression levels of important genes.

Overall, this study has many strengths, is very clearly communicated, and has no substantial weaknesses that I can point to.

One question that emerges for me is whether these findings apply broadly. In other words, I wonder whether gene expression levels are predictive of other phenotypes in other organisms. I think this question has largely been explored in microbes, where some studies (PMID: 17959824) but not others (PMID: 38895328) found that differences in gene expression were predictive of phenotypes like growth rate. Microbes are not the focus here, and instead, the discussion is mainly focused on using gene expression to predict health and disease phenotypes in humans. This feels a little complicated since humans have so many different tissues. Perhaps an area where this approach might be useful is in examining infectious single-cell populations (bacteria, tumors, fungi). But I suppose this idea might still work in humans, assuming the authors are thinking about targeting specific tissues for RNAseq.

In sum, this is a great paper that really got me thinking about the predictive power of gene expression and where/when it could inform about (health-related) phenotypes.

Comments on revisions: No additional comments

---

## [Referee Report · Reviewer #3 (Public review)]

Summary:

Webster et al. sought to understand if phenotypic variation in the absence of genetic variation can be predicted by variation in gene expression. To this end they quantified two reproductive traits, the onset of egg laying and early brood size in cohorts of genetically identical nematodes exposed to alternative ancestral (two maternal ages) and same generation life histories (either constant 20 ºC temperature or 8-hour temperature shift to 25 ºC upon hatching) in a two-factor design; then, they profiled genome-wide gene expression in each individual.

Using multiple statistical and machine learning approaches, they showed that, at least for early brood size, phenotypic variation can be quite well predicted by molecular variation, beyond what can be predicted by life history alone.

Moreover, they provide some evidence that expression variation in some genes might be causally linked to phenotypic variation.

Strengths:

Cleverly designed and carefully performed experiments that provide high-quality datasets useful for the community.

Good evidence that phenotypic variation can be predicted by molecular variation.

Weaknesses:

What drives the molecular variation that impacts phenotypic variation remains unknown. While the authors show that variation in expression of some genes might indeed be causal, it is still not clear how much of the molecular variation is a cause rather than a consequence of phenotypic variation.

Comments on revisions: I have no more comments for the authors

---

## [Author Response]

The following is the authors’ response to the original reviews

**Reviewer #1 (Public review):**
Summary:Genome-wide association studies have been an important approach to identifying the genetic basis of human traits and diseases. Despite their successes, for many traits, a substantial amount of variation cannot be explained by genetic factors, indicating that environmental variation and individual 'noise' (stochastic differences as well as unaccounted for environmental variation) also play important roles. The authors' goal was to address whether gene expression variation in genetically identical individuals, driven by historical environmental differences and 'noise', could be used to predict reproductive trait differences.Strengths:To address this question, the authors took advantage of genetically identical *C. elegans* individuals to transcriptionally profile 180 adult hermaphrodite individuals that were also measured for two reproductive traits. A major strength of the paper is its experimental design. While experimenters aim to control the environment that each worm experiences, it is known that there are small differences that each worm experiences even when they are grown together on the same agar plate - e.g. the age of their mother, their temperature, the amount of food they eat, and the oxygen and carbon dioxide levels depending on where they roam on the plate. Instead of neglecting this unknown variation, the authors design the experiment up front to create two differences in the historical environment experienced by each worm: (1) the age of its mother and (2) 8 8-hour temperature difference, either 20 or 25 {degree sign}C. This helped the authors interpret the gene expression differences and trait expression differences that they observed.Using two statistical models, the authors measured the association of gene expression for 8824 genes with the two reproductive traits, considering both the level of expression and the historical environment experienced by each worm. Their data supports several conclusions. They convincingly show that gene expression differences are useful for predicting reproductive trait differences, predicting ~25-50% of the trait differences depending on the trait. Using RNAi, they also show that the genes they identify play a causal role in trait differences. Finally, they demonstrate an association with trait variation and the H3K27 trimethylation mark, suggesting that chromatin structure can be an important causal determinant of gene expression and trait variation.Overall, this work supports the use of gene expression data as an important intermediate for understanding complex traits. This approach is also useful as a starting point for other labs in studying their trait of interest.

We thank the reviewer for their thorough articulation of the strengths of our study.

Weaknesses:There are no major weaknesses that I have noted. Some important limitations of the work (that I believe the authors would agree with) are worth highlighting, however:(1) A large remaining question in the field of complex traits remains in splitting the role of non-genetic factors between environmental variation and stochastic noise. It is still an open question which role each of these factors plays in controlling the gene expression differences they measured between the individual worms.

Yes, we agree that this is a major question in the field. In our study, we parse out differences driven between known historical environmental factors and unknown factors, but the ‘unknown factors’ could encompass both unknown environmental factors and stochastic noise.

(2) The ability of the authors to use gene expression to predict trait variation was strikingly different between the two traits they measured. For the early brood trait, 448 genes were statistically linked to the trait difference, while for egg-laying onset, only 11 genes were found. Similarly, the total R2 in the test set was ~50% vs. 25%. It is unclear why the differences occur, but this somewhat limits the generalizability of this approach to other traits.

We agree that the difference in predictability between the two traits is interesting. A previous study from the Phillips lab measured developmental rate and fertility across *Caenorhabditis* species and parsed sources of variation (1). Results indicated that 83.3% of variation in developmental rate was explained by genetic variation, while only 4.8% was explained by individual variation. In contrast, for fertility, 63.3% of variation was driven by genetic variation and 23.3% was explained by individual variation. Our results, of course, focus only on predicting the individual differences, but not genetic differences, for these two traits using gene expression data. Considering both sets of results, one hypothesis is that we have more power to explain nongenetic phenotypic differences with molecular data if the trait is less heritable, which is something that could be formally interrogated with more traits across more strains.

(3) For technical reasons, this approach was limited to whole worm transcription. The role of tissue and celltype expression differences is important to the field, so this limitation is important.

We agree with this assessment, and it is something we hope to address with future work.

**Reviewer #2 (Public review):**
Summary:This paper measures associations between RNA transcript levels and important reproductive traits in the model organism *C. elegans*. The authors go beyond determining which gene expression differences underlie reproductive traits, but also (1) build a model that predicts these traits based on gene expression and (2) perform experiments to confirm that some transcript levels indeed affect reproductive traits. The clever study design allows the authors to determine which transcript levels impact reproductive traits, and also which transcriptional differences are driven by stochastic vs environmental differences. In sum, this is a rather comprehensive study that highlights the power of gene expression as a driver of phenotype, and also teases apart the various factors that affect the expression levels of important genes.Strengths:Overall, this study has many strengths, is very clearly communicated, and has no substantial weaknesses that I can point to. One question that emerges for me is about the extent to which these findings apply broadly. In other words, I wonder whether gene expression levels are predictive of other phenotypes in other organisms. Ithink this question has largely been explored in microbes, where some studies (PMID: 17959824) but not others (PMID: 38895328) find that differences in gene expression are predictive of phenotypes like growth rate. Microbes are not the primary focus here, and instead, the discussion is mainly focused on using gene expression to predict health and disease phenotypes in humans. This feels a little complicated since humans have so many different tissues. Perhaps an area where this approach might be useful is in examining infectious single-cell populations (bacteria, tumors, fungi). But I suppose this idea might still work in humans, assuming the authors are thinking about targeting specific tissues for RNAseq.In sum, this is a great paper that really got me thinking about the predictive power of gene expression and where/when it could inform about (health-related) phenotypes.

We thank the reviewer for recognizing the strengths of our study. We are also interested in determining the extent to which predictive gene expression differences operate in specific tissues.

**Reviewer #3 (Public review):**
Summary:Webster et al. sought to understand if phenotypic variation in the absence of genetic variation can be predicted by variation in gene expression. To this end they quantified two reproductive traits, the onset of egg laying and early brood size in cohorts of genetically identical nematodes exposed to alternative ancestral (two maternal ages) and same generation life histories (either constant 20C temperature or 8-hour temperature shift to 25C upon hatching) in a two-factor design; then they profiled genome-wide gene expression in each individual.Using multiple statistical and machine learning approaches, they showed that, at least for early brood size, phenotypic variation can be quite well predicted by molecular variation, beyond what can be predicted by life history alone.Moreover, they provide some evidence that expression variation in some genes might be causally linked to phenotypic variation.Strengths:(1) Cleverly designed and carefully performed experiments that provide high-quality datasets useful for the community.(2) Good evidence that phenotypic variation can be predicted by molecular variation.

We thank the reviewer for recognizing the strengths of our study.

Weaknesses:What drives the molecular variation that impacts phenotypic variation remains unknown. While the authors show that variation in expression of some genes might indeed be causal, it is still not clear how much of the molecular variation is a cause rather than a consequence of phenotypic variation.

We agree that the drivers of molecular variation remain unknown. While we addressed one potential candidate (histone modifications), there is much to be done in this area of research. We agree that, while some gene expression differences cause phenotypic changes, other gene expression differences could in principle be downstream of phenotypic differences.

**Recommendations for the authors:**

**Reviewer #1 (Recommendations for the authors):**
I have a number of suggestions that I believe will improve the Methods section.(1) Strain N2-PD1073 will probably be confusing to some readers. I recommend spelling out that this is the Phillips lab version of N2.

Thank you for this suggestion; we have added additional explanation of this strain in the Methods.

(2) I found the details of the experimental design confusing, and I believe a supplemental figure will help. I have listed the following points that could be clarified:a. What were the biological replicates? How many worms per replicate?

Biological replicates were defined as experiments set up on different days (in this case, all biological replicates were at least a week apart), and the biological replicate of each worm can be found in Supplementary File 1 on the Phenotypic Data tab.

b. I believe that embryos and L4s were picked to create different aged P0s, and eggs and L4s were picked to separate plates? Is this correct?

Yes, this is correct.

c. What was the spread in the embryo age?

We assume this is asking about the age of the F1 embryos, and these were laid over the course of a 2-hour window.

d. While the age of the parents is different, there are also features about their growth plates that will be impacted by the experimental design. For example, their pheromone exposure is different due to the role that age plays in the combination of ascarosides that are released. It is worth noting as my reading of the paper makes it seem that parental age is the only thing that matters.

The parents (P0) of different ages likely have differential ascaroside exposure because they are in the vicinity of other similarly aged worms, but the F1 progeny were exposed to their parents for only the 2-hour egg-laying window, in an attempt to minimize this type of effect as much as possible.

e. Were incubators used for each temperature?

Yes.

f. In line 443, why approximately for the 18 hours? How much spread?

The approximation was based on the time interval between the 2-hour egg-laying window on Day 4 and the temperature shift on Day 5 the following morning. The timing was within 30 minutes of 18 hours either direction.

g. In line 444, "continually left" is confusing. Does this mean left in the original incubator?

Yes, this means left in the incubator while the worms shifted to 25°C were moved. To avoid confusion, we re-worded this to state they “remained at 20°C while the other half were shifted to 25°C”.

h. In line 445, "all worms remained at 20 {degree sign}C" was confusing to me as to what it indicated. I assume, unless otherwise noted, the animals would not be moved to a new temperature.

This was an attempt to avoid confusion and emphasize that all worms were experiencing the same conditions for this part of the experiment.

i. What size plates were the worms singled onto?

They were singled onto 6-cm plates.

j. If a figure were to be made, having two timelines (with respect to the P0 and F1) might be useful.

We believe the methods should be sufficient for someone who hopes to repeat the experiment, and we believe the schematic in Figure 1A labeling P0 and F1 generations is sufficient to illustrate the key features of the experimental design.

k. Not all eggs that are laid end up hatching. Are these censored from the number of progeny calculations?

Yes, only progeny that hatched and developed were counted for early brood.

(3) For the lysis, was the second transfer to dH20 also a wash step?

Yes.

(4) What was used for the Elution buffer?

We used elution buffer consisting of 10 mM Tris, 0.1 mM EDTA. We have added this to the “Cell lysate generation” section of the methods

(5) The company that produced the KAPA mRNA-seq prep kit should be listed.

We added that the kit was from Roche Sequencing Solutions.

(6) For the GO analysis - one potential issue is that the set of 8824 genes might also be restricted to specific GO categories. Was this controlled for?

We originally did not explicitly control for this and used the default enrichGO settings with OrgDB = org.Ce.eg.db as the background set for *C. elegans*. We have now repeated the analysis with the “universe” set to the 8824-gene background set. This did not qualitatively change the significant GO terms, though some have slightly higher or lower p-values. For comparison purposes, we have added the background-corrected sets to the GO_Terms tab of Supplementary File 1 with each of the three main gene groups appended with “BackgroundOf8824”.

**Reviewer #2 (Recommendations for the authors):**
(1) The abstract, introduction, and experimental design are well thought through and very clear.

Thank you.

(2) Figure 1B could use a clearer or more intuitive label on the horizontal axis. The two examples help. Maybe the genes (points) on the left side should be blue to match Figure 1C, where the genes with a negative correlation are in the blue cluster.

Thank you for these suggestions. We re-labeled the x-axis as “Slope of early brood vs. gene expression (normalized by CPM)”, which we hope gives readers a better intuition of what the coefficient from the model is measuring. We also re-colored the points previously colored red in Figure 1B to be color-coded depending on the direction of association to match Figure 1C, so these points are now color-coded as pink and purple.

(3) If red/blue are pos/neg correlated genes in 1C, perhaps different colors should be used to label ELO and brood in Figures 2 and 3. Green/purple?

We appreciate this point, but since we ended up using the cluster colors of pink and purple in Figure 1, we opted to leave Figures 2 and 3 alone with the early brood and ELO colorcoding of red and blue.

(4) I am unfamiliar with this type of beta values, but I thought the explanation and figure were very clear. It could be helpful to bold beta1 and beta2 in the top panels of Figure 2, so the readers are not searching around for those among all the other betas. It could also be helpful to add an English phrase to the vertical axes inFigures 2C and 2D, in addition to the beta1 and beta2. Something like "overall effect (beta1)" and"environment-controlled effect (beta2)". Or maybe "effect of environment + stochastic expression differences(beta1)" and "effect of stochastic expression differences alone (beta2)". I guess those are probably too big to fit on the figure, but it might be nice to have a label somewhere on this figure connecting them to the key thing you are trying to measure - the effect of gene expression and environment.

Thank you for these suggestions. We increased the font sizes and bolded β1 and β2 in Figure 2A-B. In Figure 2C-D, we added a parenthetical under β1 to say “(env + noise)” and β2 to say “(noise)”. We agree that this should give the reader more intuition about what the β values are measuring.

**Reviewer #3 (Recommendations for the authors):**
The authors collected individuals 24 hours after the onset of egg laying for transcriptomic profiling. This is a well-designed experiment to control for the physiological age of the germline. However, this does not properly control for somatic physiological age. Somatic age can be partially uncoupled from germline age across individuals, and indeed, this can be due to differences in maternal age (Perez et al, 2017). This is because maternal age is associated with increased pheromone exposure (unless you properly controlled for it by moving worms to fresh plates), which causes a germline-specific developmental delay in the progeny, resulting in a delayed onset of egg production compared to somatic development (Perez et al. 2021). You control for germline age, therefore, it is likely that the progeny of day 1 mothers are actually somatically older than the progeny of day 3 mothers. This would predict that many genes identified in these analyses might just be somatic genes that increase or decrease their expression during the young adult stage.For example, the abundance of collagen genes among the genes negatively associated (including col-20, which is the gene most significantly associated with early brood) is a big red flag, as collagen genes are known to be changing dynamically with age. If variation in somatic vs germline age is indeed what is driving the expression variation of these genes, then the expectation is that their expression should decrease with age. Vice versa, genes positively associated with early brood that are simply explained by age should be increasing. So I would suggest that the authors first check this using time series transcriptomic data covering the young adult stage they profiled. If this is indeed the case, I would then suggest using RAPToR (https://github.com/LBMC/RAPToR), a method that, using reference time series data, can estimate physiological age (including tissue-specific one) from gene expression. Using this method they can estimate the somatic physiological age of their samples, quantify the extent of variation in somatic age across individuals, quantify how much of the observed differences in expressions are explained just by differences in somatic age and correct for them during their transcriptomic analysis using the estimated soma age as a covariate (https://github.com/LBMC/RAPToR/blob/master/vignettes/RAPToR-DEcorrection-pdf.pdf).This should help enrich a molecular variation that is not simply driven by hidden differences between somatic and germline age.

To first address some of the experimental details mentioned for our paper, parents were indeed moved to fresh plates where they were allowed to lay embryos for two hours and then removed. Thus, we believe this minimizes the effects of ascarosides as much as possible within our design. As shown in the paper, we also identified genes that were not driven by parental age and for all genes quantified to what extent each gene’s association was driven by parental age. Thus, it is unlikely that differences in somatic and germline age is the sole explanatory factor, even if it plays some role. We also note that we accounted for egg-laying onset timing in our experimental design, and early brood was calculated as the number of progeny laid in the first 24 hours of egg-laying, where egg-laying onset was scored for each individual worm to the hour. The plot of each worm’s ELO and early brood traits is in Figure S1. Nonetheless, we read the RAPToR paper with interest, as we highlighted in the paper that germline genes tend to be positively associated with early brood while somatic genes tend to be negatively associated. While the RAPToR paper discusses using tissue-specific gene sets to stage genetically diverse *C. elegans* RILs, the RAPToR reference itself was not built using gene expression data acquired from different *C. elegans* tissues and is based on whole worms, typically collected in bulk. I.e., age estimates in RILs differ depending on whether germline or somatic gene sets are used to estimate age when the the aging clock is based on N2 samples. Thus, it is unclear whether such an approach would work similarly to estimate age in single worm N2 samples. In addition, from what we can tell, the RAPToR R package appears to implement the overall age estimate, rather than using the tissue-specific gene sets used for RILs in the paper. Because RAPToR would be estimating the overall age of our samples using a reference that is based on fewer samples than we collected here, and because we already know the overall age of our samples measured using standard approaches, we believe that estimating the age with the package would not give very much additional insight.

Bonferroni correction:First, I think there is some confusion in how the author report their p-values: I don't think the authors are using a cut-off of Bonferroni corrected p-value of 5.7 x 10-6 (it wouldn't make sense). It's more likely that they are using a Bonferroni corrected p of 0.05 or 0.1, which corresponds to a nominal p value of 5.7 x 10-6, am I right?

Yes, we used a nominal p-value of 5.7 x 10-6 to correspond to a Bonferroni-corrected p-value of 0.05, calculated as 0.05/8824. We have re-worded this wherever Bonferroni correction was mentioned.

Second, Bonferroni is an overly stringent correction method that has now been substituted by the more powerful Benjamini Hochberg method to control the false discovery rate. Using this might help find more genes and better characterize the molecular variation, especially the one associated with ELO?

We agree that Bonferroni is quite stringent and because we were focused on identifying true positives, we may have some false negatives. Because all nominal p-values are included in the supplement, it is straightforward for an interested reader to search the data to determine if a gene is significant at any other threshold.

Minor comments:(1) "In our experiment, isogenic adult worms in a common environment (with distinct historical environments) exhibited a range of both ELO and early brood trait values (Fig S1A)" I think this and the figure is not really needed, Figure S1B is already enough to show the range of the phenotypes and how much variation is driven by the life history traits.

We agree that the information in S1A is also included in S1B, but we think it is a little more straightforward if one is primarily interested in viewing the distribution for a single trait.

(2) Line 105 It should be Figure S2, not S3.

Thank you for catching this mistake.

(3) Gene Ontology on positive and negatively associated genes together: what about splitting the positive and negative?

We have added a split of positive and negative GO terms to the GO_Terms tab of Supplement File 1. Broadly speaking, the most enriched positively associated genes have many of the same GO terms found on the combined list that are germline related (e.g., involved in oogenesis and gamete generation), whereas the most enriched negatively associated genes have GO terms found on the combined list that are related to somatic tissues (e.g., actin cytoskeleton organization, muscle cell development). This is consistent with the pattern we see for somatic and germline genes shown in Figure 4.

(4) A lot of muscle-related GOs, can you elaborate on that?

Yes, there are several muscle-related GOs in addition to germline and epidermis. While we do not know exactly why from a mechanistic perspective these muscle-related terms are enriched, it may be important to note that many of these terms have highly overlapping sets of genes which are listed in Supplementary File 1. For example, “muscle system process” and “muscle contraction” have the exact same set of 15 genes causing the term to be significantly enriched. Thus, we tend to not interpret having many GO terms on a given tissue as indicating that the tissue is more important than others for a given biological process. While it is clear there are genes related to muscle that are associated with early brood, it is not yet clear that the tissue is more important than others.

(5) "consistent with maternal age affecting mitochondrial gene expression in progeny " - has this been previously reported?

We do not believe this particular observation has been reported. It is important to note that these genes are involved in mitochondrial processes, but are expressed from the nuclear rather than mitochondrial genome. We re-worded the quoted portion of the sentence to say “consistent with parental age affecting mitochondria-related gene expression in progeny”.

(6) PCA: "Therefore, the optimal number of PCs occurs at the inflection points of the graph, which is after only7 PCs for early brood (R2 of 0.55) but 28 PCs for ELO (R2 of 0.56)."Not clear how this is determined: just graphically? If yes, there are several inflection points in the plot. How did you choose which one to consider? Also, a smaller component is not necessarily less predictive of phenotypic variation (as you can see from the graph), so instead of subsequently adding components based on the variance, they explain the transcriptomic data, you might add them based on the variance they explain in the phenotypic data? To this end, have you tried partial least square regression instead of PCA? This should give gene expression components that are ranked based on how much phenotypic variance they explain.

Thank you for this thoughtful comment. We agree that, unlike for Figure 3B, there is some interpretation involved on how many PCs is optimal because additional variance explained with each PC is not strictly decreasing beyond a certain number of PCs. Our assessment was therefore made both graphically and by looking at the additional variance explained with each additional PC. For example, for early brood, there was no PC after PC7 that added more than 0.04 to the R2. We could also have plotted early brood and ELO separately and had a different ordering of PCs on the x-axis. By plotting the data this way, we emphasized that the factors that explain the most variation in the gene expression data typically explain most variation in the phenotypic data.

(7) The fact that there are 7 PC of molecular variation that explain early brood is interesting. I think the authors can analyze this further. For example, could you perform separate GO enrichment for each component that explains a sizable amount of phenotypic variance? Same for the ELO.

Because each gene has a PC loading in for each PC, and each PC lacks the explanatory power of combined PCs, we believe doing GO Terms on the list of genes that contribute most to each PC is of minimal utility. The power of the PCA prediction approach is that it uses the entire transcriptome, but the other side of the coin is that it is perhaps less useful to do a gene-bygene based analysis with PCA. This is why we separately performed individual gene associations and 10-gene predictive analyses. However, we have added the PC loadings for all genes and all PCs to Supplementary File 1.

(8) Avoid acronyms when possible (i.e. ELO in figures and figure legends could be spelled out to improve readability).

We appreciate this point, but because we introduced the acronym both in Figure 1 and the text and use it frequently, we believe the reader will understand this acronym. Because it is sometimes needed (especially in dense figures), we think it is best to use it consistently throughout the paper.

(9) Multiple regression: I see the most selected gene is col-20, which is also the most significantly differentially expressed from the linear mixed model (LMM). But what is the overlap between the top 300 genes in Figure 3F and the 448 identified by the LMM? And how much is the overlap in GO enrichment?

Genes that showed up in at least 4 out of 500 iterations were selected more often than expected by chance, which includes 246 genes (as indicated by the red line in Figure 3F). Of these genes, 66 genes (27%) are found in the set of 448 early brood genes. The proportion of overlap increases as the number of iterations required to consider a gene predictive increases, e.g., 34% of genes found in 5 of 500 iterations and 59% of genes found in 10 of 500 iterations overlap with the 448 early brood genes. However, likely because of the approach to identify groups of 10 genes that are predictive, we do not find significant GO terms among the 246 genes identified with this approach after multiple test correction. We think this makes sense because the LMM identifies genes that are individually associated with early brood, whereas each subsequent gene included in multiple regression affects early brood after controlling for all previous genes. These additional genes added to the multiple regression are unlikely to have similar patterns as genes that are individually correlated with early brood.

(10) Elastic nets: prediction power is similar or better than multiple regression, but what is the overlap between genes selected by the elastic net (not presented if I am not mistaken) and multiple regression and the linear mixed model?

For the elastic net models, we used a leave-one-out cross validation approach, meaning there were separate models fit by leaving out the trait data for each worm, training a model using the trait data and transcriptomic data for the other worms, and using the transcriptomic data of the remaining worm to predict the trait data. By repeating this for each worm, the regressions shown in the paper were obtained. Each of these models therefore has its own set of genes. Of the 180 models for early brood, the median model selects 83 genes (range from 72 to 114 genes). Across all models, 217 genes were selected at least once. Interestingly, there was a clear bimodal distribution in terms of how many models a given gene was selected for: 68 genes were selected in over 160 out of 180 models, while 114 genes were selected in fewer than 20 models (and 45 genes were selected only once). Therefore, we consider the set of 68 genes as highly robustly selected, since they were selected in the vast majority of models. This set of 68 exhibits substantial overlap with both the set of 448 early brood-associated genes (43 genes or 63% overlap) and the multiple regression set of 246 genes (54 genes or 79% overlap). For ELO, the median model selected 136 genes (range of 96 to 249 genes) and a total of 514 genes were selected at least once. The distribution for ELO was also bimodal with 78 genes selected over 160 times and 255 genes selected fewer than 20 times. This set of 78 included 6 of the 11 significant ELO genes identified in the LMM. We have added tabs to Supplementary File 1 that include the list of genes selected for the elastic net models as well as a count of how many times they were selected out of 180 models.

(11) In other words, do these different approaches yield similar sets of genes, or are there some differences?

In the end, which approach is actually giving the best predictive power? From the perspective of R2, both the multiple regression and elastic net models are similarly predictive for early brood, but elastic net is more predictive for ELO. However, in presenting multiple approaches, part of our goal was identifying predictive genes that could be considered the ‘best’ in different contexts. The multiple regression was set to identify exactly 10 genes, whereas the elastic net model determined the optimal number of genes to include, which was always over 70 genes. Thus, the elastic net model is likely better if one has gene expression data for the entire transcriptome, whereas the multiple regression genes are likely more useful if one were to use reporters or qRTPCR to measure a more limited number of genes.

(12) Line 252: "Within this curated set, genes causally affected early brood in 5 of 7 cases compared to empty vector (Figure 4A).

" It seems to me 4 out of 7 from Figure 4A. In Figure 4A the five genes are (1) *cin-4*, (2) *puf5; puf-7*, (3) *eef-1A.2*, (4) C34C12.8, and (5) *tir-1*. We did not count *nex-2* (p = 0.10) or *gly-13* (p = 0.07), and empty vector is the control.

(13) Do puf-5 and -7 affect total brood size or only early brood size? Not clear. What's the effect of single puf-5 and puf-7 RNAi on brood?

We only measured early brood in this paper, but a previous report found that *puf-5* and *puf-7* act redundantly to affect oogenesis, and RNAi is only effective if both are knocked down together(2). We performed pilot experiments to confirm that this was the case in our hands as well.

(14) To truly understand if the noise in expression of Puf-5 and /or -7 really causes some of the observed difference in early brood, could the author use a reporter and dose response RNAi to reduce the level of puf-5/7 to match the lower physiological noise range and observe if the magnitude of the reduction of early brood by the right amount of RNAi indeed matches the observed physiological "noise" effect of puf-5/7 on early brood?

We agree that it would be interesting to do the dose response of RNAi, measure early brood, and get a readout of mRNA levels to determine the true extent of gene knockdown in each worm (since RNAi can be noisy) and whether this corresponds to early brood when the knockdown is at physiological levels. While we believe we have shown that a dose response of gene knockdown results in a dose response of early brood, this additional analysis would be of interest for future experiments.

(15) Regulated soma genes (enriched in H3K27me3) are negatively correlated with early brood. What would be the mechanism there? As mentioned before, it is more likely that these genes are just indicative of variation in somatic vs germline age (maybe due to latent differences in parental perception of pheromone).

We can think of a few potential mechanisms/explanations, but at this point we do not have a decisive answer. Regulated somatic genes marked with H3K27me3 (facultative heterochromatin) are expressed in particular tissues and/or at particular times in development. In this study and others, genes marked with H3K27me3 exhibit more gene expression noise than genes with other marks. This could suggest that there are negative consequences for the animal if genes are expressed at higher levels at the wrong time or place, and one interpretation of the negative association is that higher expressed somatic genes results in lower fitness (where early brood is a proxy for fitness). Another related interpretation is that there are tradeoffs between somatic and germline development and each individual animal lands somewhere on a continuum between prioritizing germline or somatic development, where prioritizing somatic integrity (e.g. higher expression of somatic genes) comes at a cost to the germline resulting in fewer progeny. Additional experiments, including measurements of histone marks in worms measured for the early brood trait, would likely be required to more decisively answer this question.

(16) Line 151: "Among significant genes for both traits, β2 values were consistently lower than β1 (Figures 2CD), suggesting some of the total effect size was driven by environmental history rather than pure noise".

We are interpreting this quote as part of point 17 below.

(17) It looks like most of the genes associated with phenotypes from the univariate model have a decreased effect once you account for life history, but have you checked for cases where the life history actually masks the effect of a gene? In other words, do you have cases where the effect of gene expression on a phenotype is only (or more) significant after you account for the effect of life history (β2 values higher than β1)?

This is a good question and one that we did not explicitly address in the paper because we focused on beta values for genes that were significant in the univariate analysis. Indeed, for the sets of 448 early brood genes ad 11 ELO genes, there are no genes for which β2 is larger than β1. In looking at the larger dataset of 8824 genes, with a Bonferroni-corrected p-value of 0.05, there are 306 genes with a significant β2 for early brood. The majority (157 genes) overlap with the 448 genes significant in the univariate analysis and do not have a higher β2 than β1. Of the remaining genes, 72 of these have a larger β2 than β1. However, in most cases, this difference is relatively small (median difference of 0.025) and likely insignificant. There are only three genes in which β1 is not nominally significant, and these are the three genes with the largest difference between β1 and β2 with β2 being larger (differences of 0.166, 0.155, and 0.12). In contrast, the *median* difference between β1 and β2 the 448 genes (in which β1 is larger) is 0.17, highlighting the most extreme examples of β2 > β1 are smaller in magnitude than the typical case of β1 > β2. For ELO, there are no notable cases where β2 > β1. There are eight genes with a significant β2 value, and all of these have a β1 value that is nominally significant. Therefore, while this phenomenon does occur, we find it to be relatively rare overall. For completeness, we have added the β1 and β2 values for all 8824 genes as a tab in Supplementary File 1.